# Arctic lead detection using a waveform mixture algorithm from CryoSat-2 data

Sanggyun Lee[1], Hyun-Cheol Kim[2], and Jungho Im[1,3,*]

[1]School of Urban and Environmental Engineering, Ulsan National Institute of Science and Technology (UNIST), Ulsan, 44919, South Korea
[2]Unit of Arctic Sea-Ice prediction, Korea Polar Research Institute (KOPRI), Incheon, 21990, South Korea
[3]Environmental Resource Engineering, State University of New York, College of Environmental Science and Forestry, 13210, Syracuse, NY, USA

*Correspondence to*: Jungho Im (ersgis@unist.ac.kr)

**Abstract.** Arctic sea ice leads play a major role in exchanging heat and momentum between the Arctic atmosphere and ocean as well as in the retrieval of sea ice thickness. Although leads cover only a small portion of the Arctic Ocean, they affect the heat budget in the Arctic region considerably. In this study, we propose a waveform mixture analysis to detect leads from CryoSat-2 data, which is novel and different from the existing threshold-based lead detection methods. The waveform mixture analysis adopts the concept of spectral mixture analysis that is widely used in the field of hyperspectral image analysis. This lead detection method, based on the waveform mixture analysis, was evaluated with high resolution (250m) MODIS images and showed comparable and promising performance in detecting leads when compared to the previous methods. The robustness of the proposed approach also lies in the fact that it does not require the rescaling of parameters (i.e., stack standard deviation, stack skewness, stack kurtosis, pulse peakiness, and backscatter sigma), as it directly uses L1B waveform data unlike the existing threshold-based methods. Monthly lead fraction maps were produced by waveform mixture analysis, which show a strong inter-annual variability of recent sea ice cover during 2011-2016, excluding the summer season (i.e., June to September). We also compared the lead fraction maps to other lead fraction maps generated from previously published data sets, resulting in similar spatiotemporal patterns.

## 1 Introduction

Sea ice leads (hereafter referred to as "leads"), linearly elongated cracks in sea ice, are a common feature in the Arctic Ocean. Leads facilitate an amount of heat and moisture exchanges between the atmosphere and the ocean because of the temperature differences between them (Maykut. 1982; Perovich et al., 2011). Although leads occupy a small portion of the Arctic Ocean, there is much more heat transfer between the atmosphere and ocean through leads than sea ice (Maykut, 1978; Marcq and Weiss, 2012). Furthermore, Lüpkes et al. (2008) showed that a 1% change in sea ice concentration owing to an increase of lead fraction could increase near surface temperature in the Arctic by 3.5 K. Thus, the detection and monitoring

of leads in the Arctic Ocean are crucial because they are closely related to the Arctic heat budget and the physical interaction between the atmospheric boundary layers and sea ice in the Arctic.

Satellite sensors have been the most efficient way to monitor leads in the entire Arctic region since the 1990s (Key et al., 1993; Lindsay and Rothrock, 1995; Miles and Barry, 1998). Advanced Very High Resolution Radiometer (AVHRR) and Defense Meteorological Satellite Program (DMSP) satellite visible and thermal images were used to detect leads in the early 1990s. Recently, the Moderate Resolution Imaging Spectroradiometer (MODIS) Ice Surface Temperature (IST) product with 1km spatial resolution was used to detect leads to map pan-Arctic lead presence (Willmes and Heinemann, 2015; Willmes and Heinemann, 2016). They mitigated cloud interference using a fuzzy cloud artefact filter and investigated lead dynamics based on a comparison between pan-Arctic lead maps and the characteristics of the Arctic Ocean such as shear zones, bathymetry, and currents. While optical sensors have a finer spatial resolution, they are not pragmatic in the dark regions during polar nights (from December to February). In addition, leads are easily contaminated by clouds. Microwave instruments such as passive microwave sensors and altimeters have been used to detect leads and to produce lead fractions. Röhrs and Kaleschke (2012) utilized the polarization ratio of the Advanced Microwave Scanning Radiometer for EOS (AMSR-E) channels and retrieved daily thin ice concentration. With the help of the thin ice concentration, lead orientations and frequencies were derived using an image analysis technique (i.e., Hough transform) (Bröhan and Kaleschke, 2014). Airborne and spaceborne radar altimeters can detect leads as well. Zygmuntowska et al. (2013) used Airborne Synthetic Aperture and Interferometric Radar Altimeter System (ASIRAS), similar to CryoSat-2, to identify leads based on waveform characteristics and a Bayesian classifier. Zakharova et al. (2015) and Wernecke and Kaleschke (2015) used the spaceborne altimeters Satellite with Argos and Altika (SARAL) and CryoSat-2 to identify leads, respectively. While Zakharova et al. (2015) applied simple thresholds to identify leads along with Satellite with Argos and Altika (SARAL/Altika) tracks and estimated regional lead fractions, Wernecke and Kaleschke (2015) optimized thresholds to detect leads and produced pan-Arctic lead fraction maps using CryoSat-2 with an analysis of lead width, and sea surface height.

Spectral mixture analysis based on the assumption that the spectra measured by sensors for a pixel are a linear combination of the spectra for all components within the pixel (Keshava and Mustard, 2002) was first applied to the altimetry research field in the Polar Region by Chase and Hoyer (1990). They estimated sea ice type and concentration using spectral mixture analysis based on Geosat waveforms. However, Geosat with a relatively small number of bins and coarser spatial resolution is not sufficient to detect small leads in the winter (DJF) and spring seasons (MAM) in the Arctic. In this study, we adopted the linear mixture analysis concept to waveforms from Synthetic Aperture Interferometric Radar Altimeter (SIRAL), CryoSat-2, to identify leads and produce monthly pan-Arctic lead fractions from January to May and October to December between 2011 and 2016. Waveform endmembers are crucial to implement spectral mixture analysis (Fig. 1). The N-FINDR (N-finder) algorithm was used to select waveform endmembers from extracted waveforms by Decision tree (DT) from Lee et al. (2016), which avoids the subjective selection of endmembers. The detected leads were visually evaluated with MODIS images (at 250 m resolution) and compared with other thresholds based lead detection

methods. The lead detection of waveform mixture analysis is not easily affected by the update of the CryoSat-2 baseline, which is novel and different from previous threshold based lead detection methods. The main objectives of this study are to 1) develop a novel lead detection method based on waveform mixture analysis, 2) compute recent pan-Arctic lead fractions, and 3) briefly examine the relationship between Arctic lead fraction and thermodynamics and ice dynamics.

## 2. Data

### 2. 1 CryoSat-2

CryoSat-2, the carrying Synthetic Aperture Interferometric Radar Altimeter (SIRAL) was launched in September 2010 by the European Space Agency (ESA). CryoSat-2 is a satellite dedicated to Polar research. SIRAL is a radar altimeter with a central frequency of 13.575 GHz ($K_u$-band) and a bandwidth of 320 MHz. CryoSat-2 takes an advantage of SIRAL to detect smaller leads with an efficient use of the instrument's energy compared to the previous radar altimeter missions such as GeoSat and Jason (Wingham et al., 2006). In this study, we used Synthetic Aperture Radar (SAR) mode, mainly operating on sea ice regions; and SAR Interferometric (SIN) mode, mainly operating on steep regions such as on the margin of an ice shelf and ice sheet of level 1b baseline C data. The SAR and SIN modes have 256 and 1024 range bins, respectively (Scagliola, 2014). The period of CryoSat-2 level 1b baseline C data in this study is in Jan. – May, Oct. – Dec. 2011-2016.

CryoSat-2 transmits bursts of radar pulses (i.e., 64) with high Pulse Repetition Frequency (PRF, 18.181kHz), which forms so-called Doppler beams because of the along-track movement of the satellite (Wingham et al., 2006). With the help of the high PRF, each Doppler beam is coherently correlated and pointed at the same location on the Earth surface. This is called beam stacking. Multi-looking is conducting by averaging the stacking beams to reduce speckles and thermal noises (Salvatore. 2013). Exemplary results waveforms in the L1b SAR data are shown in Fig. 1. Such waveforms represent temporal distribution of reflected power when the radar pulses reach the surface, describing a flat or rough surface. In this case, since the leading edge of each waveform starts from a different range bin, the beginning of the waveform was set at 1% of the maximum echo power (Figure 1). For a more detailed explanation about the processes to develop L1b waveform data, refer to Salvatore (2013).

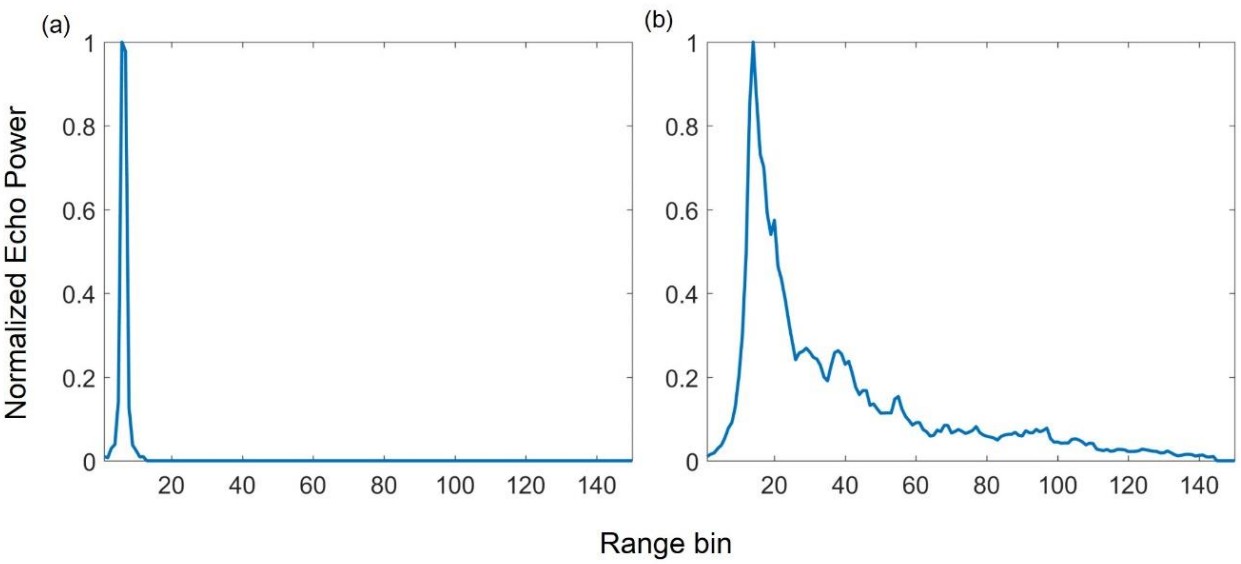

**Figure 1:** Representative waveforms of (a) leads and (b) sea ice over the Arctic Ocean selected by N-FINDR algorithm during January to May and October to December between 2011 and 2016. Refer to the methods section for N-FINDR algorithm.

## 2.2 Sea ice edge data

The European organization for the exploitation of METeorological SATellites (EUMETSAT) Ocean and Sea Ice Satellite Application Facility (OSI SAF) provides multiple sea ice products such as sea ice concentration, sea ice edge, sea ice type, sea ice emissivity, and sea ice drift. The sea ice edge product was developed using the polarization ratio of 19 GHz and 91 GHz, the spectral gradient ratio of 37 GHz and 19 GHz from Special Sensor Microwave Imager/Souder (SSMIS), and anisFMB from The Advanced Scatterometer (ASCAT) with Bayesian approach (Aaboe et al., 2016). In this study, monthly averaged sea ice edge data was used to mask out monthly lead fraction maps. The open ice cover in the sea ice edge product was regarded as an open ocean.

## 2.3 Monthly lead fraction maps

Lead fraction maps produced from previous studies (Röhrs and Kaleschke, 2012; Wernecke and Kaleschke, 2015; Willmes and Heinemann, 2016) were used to compare to the lead fraction maps generated using the proposed waveform mixture analysis in this study. Röhrs and Kaleschke (2012) produced daily thin ice concentration maps using AMSR-E data with a 6.25 km grid, which can detect leads that are wider (i.e., width) than 3 km. The daily thin ice concentration that was over 0.5 (i.e., 50%) was considered to be a lead and binary daily lead maps were averaged to properly compare other monthly lead fraction maps. A threshold optimization based lead detection method with the CryoSat-2 was used in Wernecke and Kaleschke (2015) and monthly lead fraction maps were calculated with the grids of 99.5 km. The thin ice concentration

maps (Röhrs and Kaleschke, 2012) and the lead fraction maps using CryoSat-2 (Wernecke and Kaleschke, 2015) are available on their website (http://icdc.cen.uni-hamburg.de/1/daten/cryosphere.html). Willmes and Heinemann (2016) also produced daily lead maps over the entire Arctic Region, classifying land, cloud, sea ice, lead-artefact, and lead with the spatial resolution less than 2 km. The lead class was only considered to calculate daily binary lead fraction maps. The sum of

the lead pixels was divided by days in a month (i.e., 28, 30, or 31) to make monthly lead fraction maps. This data is available on their website (http:/dx.doi.org/10.1594/PANGAEA.854411) ). In this study, we compared the monthly lead fraction maps from January to March 2011 as AMSR-E based lead fraction maps were only available until 2011.

## 3. Methods

**3.1 Waveform mixture algorithm**

An endmember in remote sensing data represents a spectrally pure ground component in a single pixel. For example, it could be pure water, vegetation, bard ground or a soil crust pixel in remote sensing data. Endmembers play the most important role in conducting spectral mixture analysis. Linear spectral mixture analysis assumes that the spectra measured by sensors for a pixel is a linear combination of the spectra of all components within the pixel (Keshava and Mustard. 2002).

This technique is widely used to resolve spectral mixture problems in image analysis (Foody and Cox, 1994; Dengsheng et al., 2003; Changshan. 2004; Iordache et al., 2011). Spectral mixture analysis determines the fractions of the components (i.e., classes) found in mixed pixels by producing abundances of the components based on endmembers. The proposed waveform mixture algorithm adopts the concept of spectral mixture analysis. Since the waveform of altimetry within a footprint could be considered to be a mixture of leads and various types of sea ice, spectral mixture analysis can be applied in this

framework. In this study, waveforms of CryoSat-2 L1b data were used as endmembers such as the waveform of pure lead and first-year ice (FYI) (Fig. 1). The lead and ice endmembers are used as reference data for separating leads and ice. In order to successfully implement waveform mixture analysis, the proper selection of lead and ice endmembers is essential.

The basic waveform mixture model is defined as follows in equation 1.

$$Y_k = \sum_{k=1}^{K} a_{ik} E_k + r_k \tag{1}$$

where $Y_p = \{Y_1, Y_2, Y_3, …, Y_k\}$ represents waveform vectors and $k$ means a range bin in the waveform. $a_{ik}$ is an abundance fraction, which provides lead and ice proportion in terms of lead and ice endmember. $E_k$ is the endmember vector. The $r_k$ represents un-modeled residual. The equation 1 is constrained under $\sum_{k=1}^{K} a_{ik} = 1$ and $a_{ik} \geq 0$. The abundance can be derived by using a least square method to minimize the un-modeled residual ($r_k$).

Chase and Holyer (1990) were concerned by two problems with the application of spectral mixture analysis to the waveform of altimeter data. First, the waveform within a footprint may not be linearly mixed between leads and sea ice. CryoSat-2 is more sensitive to the specular reflection of leads than the diffuse reflection of sea ice when both leads and sea ice exist within the same footprint, which implies the waveform may tend to be similar to the endmember of leads (Chase and Holyer. 1990). Since CryoSat-2 data have a large number of range bins, indicating higher vertical resolution than the range bins from Geosat, they could be used to reduce the overestimation of leads. Secondly, the waveform of the altimeter (i.e., Geosat) is somewhat weighted on the centre of a footprint rather than representing an entire footprint. This could be an error source when applying spectral mixture analysis to waveform data (Chase and Holyer. 1990). However, the CryoSat-2 L1b waveform is produced by averaging more than 200 weighted waveforms with various incidence angles, which can alleviate such a problem.

## 3.2 Endmember selection

The selection of endmembers is essential in the framework of waveform mixture analysis. Among CryoSat-2 orbit files between 2011 and 2016, a total of 48 orbit files were selected to extract endmember samples by month (Jan. to May and Oct. to Dec.), which fully transverse the broad Arctic Ocean (Fig. 2). The lead and ice waveforms is extracted by using the decision trees (DT) algorithm developed for lead detection by Lee et al. (2016). DT has proven to be very effective in various remote sensing classification tasks (Kim et al., 2015; Torbick and Corbiere, 2015; Amani et al., 2017; Tadesse et al., 2017). The lead and sea ice endmembers (i.e., the most representative waveforms) are a key factor in the successful implementation of the waveform mixture algorithm. In order to avoid the subjective selection of endmembers, a number of endmember candidates were extracted by the DT algorithm (Lee et al., 2016) and the N-FINDR algorithm determined the optimum lead and ice endmembers. The N-FINDR algorithm basically uses the fact that the N spectral dimension and the N-volume (V), defined by a simplex with pure pixels, are always greater than any other combinations (Winter 1999). It operates by inflating a simplex inside of the data (endmembers), starting with any pixel set. The endmember is replaced with another endmember, and the volume is recalculated. The endmember is replaced with the spectrum of the new pixel if the volume increases. This process repeats until the volume does not increase (i.e., until there is no replacement).

$$\mathbf{E} = \begin{bmatrix} 1 & 1 & \cdots & 1 \\ \vec{e_1} & \vec{e_2} & \cdots & \vec{e_i} \end{bmatrix} \qquad (2)$$

Where $\vec{e_1}$ represents a column vector of the endmember i.

$$\mathbf{V(E)} = \left| det \begin{pmatrix} 1 & 1 & \cdots & 1 \\ \vec{e_1} & \vec{e_2} & \cdots & \vec{e_i} \end{pmatrix} \right| / (i-1)! \qquad (3)$$

The Volume (V) of the simplex containing synthetic endmember sets is proportional to the determinant. This algorithm has been widely used for automatically selecting representative endmembers (Winter, 1999; Zortea and Plaza, 2009; Erturk and plaza, 2015; Ji et al., 2015; Chi et al., 2016).

The DT model from Lee et al. (2016) was developed using data (i.e., stack standard deviation, stack skewness, stack kurtosis, pulse peakiness, and backscatter sigma-0) collected in March and April 2011-2014. Thus, the waveforms in other months and years should be compared with the waveforms in March and April 2011-2014 to identify whether the waveforms derived by the DT model during the study period are appropriate to implement the waveform mixture algorithm. Waveforms from March to April between 2011 and 2014 were compared to those from January to May, and October to December between 2011 and 2016 (not shown), resulting in little difference between them using visual analysis. This justified the use of the DT algorithm proposed by Lee et al. (2016) to extract waveform samples of leads and sea ice. The total number of sea ice and lead waveforms is 420,858 and 8,501, respectively.

The lead classification based on waveform mixture analysis was evaluated with 250 m MODIS images collected from March to May and in October. We used Earth View 250m Reflective Solar Bands Scaled Integers in MOD02QKM and adjusted the contrast to emphasize leads from sea ice in the images. It should be noted that since MODIS images with spatial resolution of 250 m were not available in January, February, November, and December due to polar nights, the evaluation with MODIS images and lead classification results based on CryoSat-2 could not be used. To secure the reliability of the comparison, the temporal difference between the MODIS images and CryoSat-2 data was always under 30 minutes.

The waveform mixture model produces abundance data (i.e., lead and sea ice abundance) at along-track points with respect to each endmember of the leads and sea ice (Fig. 3). While the lead abundances are high on the leads, the ice abundances are low on the leads, and vice versa (Fig. 3). Thresholds have to be determined to make a binary classification between leads and sea ice. Optimum thresholds to produce binary lead classification from lead and sea ice abundances were identified through an automated calibration. To implement the automated calibration, reference point data of leads and sea ice were determined by visual inspection of four MODIS images collected on 17 April 2014, 25 May 2015, 10 October 2015, and 27 March 2016. While the calibration was conducted using half of the reference data randomly selected, the validation was performed using the remaining data. The size of the leads detected by the proposed waveform mixture algorithm is at least 250m or greater because the calibration and validation processes were conducted using MODIS images with 250m spatial resolution. It should be noted that leads smaller than 250m are hardly seen in MODIS images, which implies that there is some uncertainty in the comparison of the lead detection methods for small leads. Threshold combinations from 0.2 to 0.9 with a step size of 0.01, for both lead and sea ice abundances, were tested and the one resulting in the highest accuracy was determined to be an optimum threshold combination.

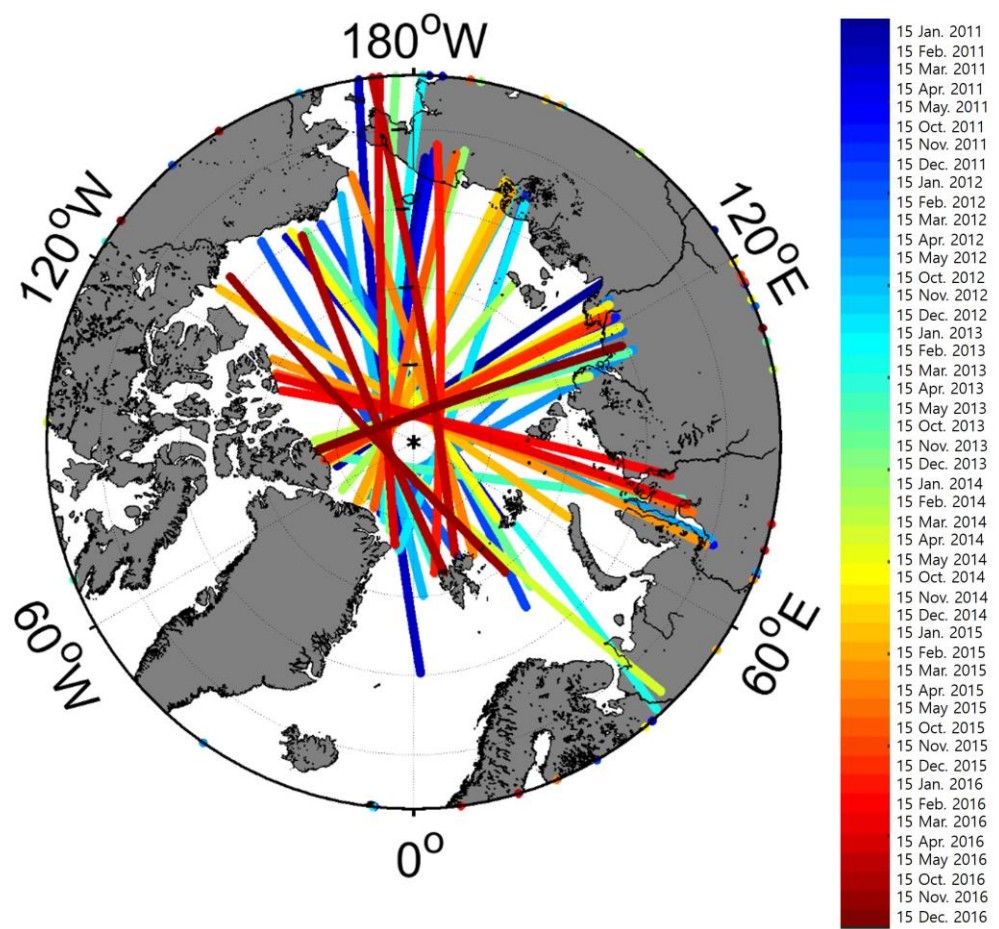

**Figure 2:** The 48 CryoSat-2 orbit files from Jan. 2011 to Dec. 2016 used for extraction endmember waveforms. The CryoSat-2 orbit files relatively cover the entire Arctic Ocean.

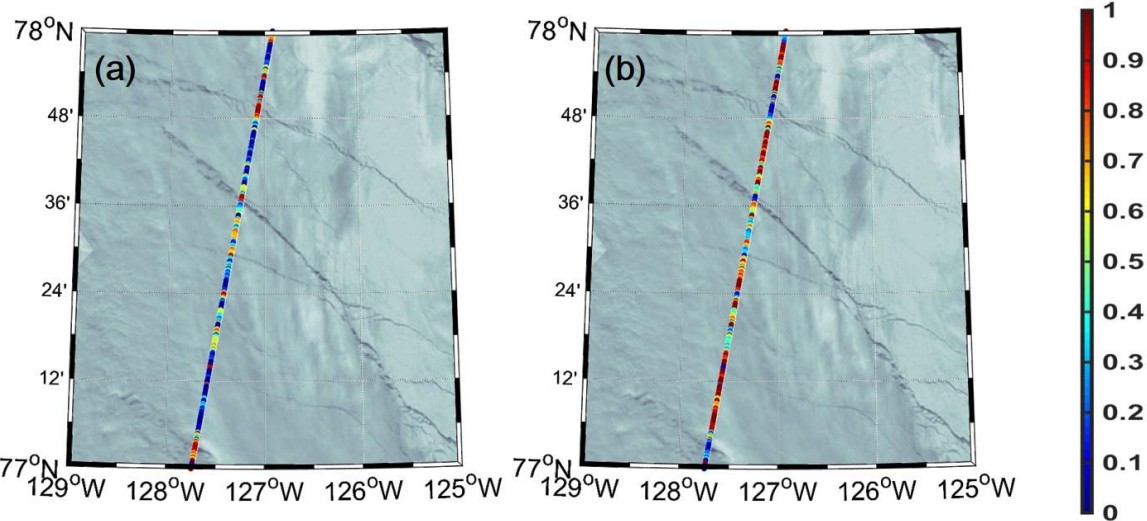

**Figure 3:** Lead and ice abundance derived by waveform mixture analysis on 10 Oct. 2015. (a) Lead abundance, (b) Ice abundance. The colour bar expresses abundances from 0 to 1.

Lead detection results were evaluated using three accuracy metrics—producer's accuracy, user's accuracy, and overall accuracy (Tab. 1). Producer's accuracy (i.e., a/(a+c) in the table), which is associated with omission errors, is calculated as the percentage of correctly classified pixels in terms of all reference samples for each class. User's accuracy (i.e., a/(a+b) in the table), which is related to commission errors, is calculated as the fraction of correctly classified pixels with regards to the pixels classified to a class. Overall accuracy (i.e., (a+d)/(a+b+c+d) in the table) is calculated as the total number of correctly classified samples divided by the total number of validation sample data. The lead and ice reference data using MODIS images and CryoSat-2 tracks were labelled through visual interpretation.

**Table 1:** Error matrix for calculation of user's, producer's and overall accuracy in terms of lead and ice classification.

| | | MODIS references | | |
|---|---|---|---|---|
| | | **Lead** | **Ice** | **Sum** |
| | **Lead** | a | b | (a+b) |
| CryoSat-2 based classification | **Ice** | c | d | (c+d) |
| | **Sum** | (a+c) | (b+d) | (a+b+c+d) |

A monthly lead fraction was derived by dividing the number of lead observations by the number of total observations within a 10 km grid in a month. It is noted that while there are more than 30 CryoSat-2 observations in the 10 km grid around the centre of the Arctic, CryoSat-2 observations less than 5 are in the 10 km grid around the coast line of Arctic Ocean. This will be dealt with in the results section with more details. It also should be noted that it is hard for the altimeter-based lead detection methods used in such as Wernecke and Kaleschke (2015) and this study to identify the propagating, opening, closing of leads because sea ice and leads generally move when the altimeters revisit a certain grid.

**3.3 Calculation of sensitivity in a 10x10 km grid**

Since each grid has a different number of CryoSat-2 observations, a sensitivity analysis was conducted in terms of the number of observations by grid. Thirty (30) percent of the lead and ice observations in 10x10 km grids was randomly permuted 50 times, and the standard deviation of the resultant lead fractions through the 50 iterations were calculated by grid. The higher the standard deviation in a grid, the more sensitive the observed lead fraction is to the number of available observations. It should be noted that the standard deviation is zero when no lead observation is found, which means lead fraction is also zero. Sensitivities were calculated from January to April 2011 because these months were used to compare the lead fractions from the proposed waveform mixture analysis to those in the existing literature.

**4 Results**

**4.1 Performance of lead classification**

Fig. 1 shows representative waveforms of leads and sea ice extracted by the N-FINDR algorithm as endmembers. The waveform of leads is dominated by specular reflection, resulting in a narrow peak curve. The representative waveform of sea ice has a wider distribution due to its rough surface when compared to that of leads. Considering different types of sea ice

such as young ice, FYI, and Multi-Year Ice (MYI), the representative waveform of sea ice is not significantly different from that of FYI based on visual inspection (Zygmuntowska et al., 2013; Ricker et al., 2015; Lee et al., 2016).

The optimum thresholds for the lead and sea ice abundances were determined to be 0.84 and 0.57 through the automated calibration, respectively. According to the thresholds, leads were identified with the conditions of lead abundance > 0.84 and sea ice abundance < 0.57. Selected examples of lead detection results based on waveform mixture analysis are presented in Fig. 4 with threshold-based lead detection results from the existing literature (Rose, 2013; Laxon et al., 2013; and Lee et al., 2016). Simple thresholding approaches based on two waveform parameters, pulse peakiness (PP) and stack standard deviation (SSD) were used in Rose (2013), Laxon et al. (2013), and Lee et al. (2016), respectively. It should be noted that since the existing methods were developed using parameters such as beam behaviour parameters and backscatter sigma-0 extracted from baseline B data, rescaling was conducted on the parameters extracted from a newly updated baseline C data for reasonable comparison. Since the contrast between the parameters of baselines B and C data is not linear, we rescaled the parameters by adding the difference of the parameters between the two baseline data to baseline C data.

Multiple lead classification methods based on CryoSat-2 data were evaluated by visual inspection with high resolution (250m) MODIS images. Leads (i.e., red dot) and sea ice (i.e., light blue dot) are distinguished, depending on the surface condition of lead and sea ice (Fig. 4). For better comparisons, a quantitative assessment is required (Fig. 4). DT from Lee et al. (2016) produced the highest overall accuracy (95.19%), followed by the waveform mixture algorithm (95%), Rose (2013) (93.26%), and Laxon et al. (2013) (91.70%). DT from Lee et al. (2016) produced the highest user's accuracy for leads, while the proposed approach produced the highest producer's accuracy for leads, which implies a slight over-detection of leads by the proposed waveform mixture algorithm. The user's accuracy for leads of Laxon et al. (2013) is the lowest, resulting in much over-detection of leads (i.e., many leads on sea ice; Fig. 4). Similarly, the user's accuracy for ice of Rose (2013) is lower than that of the proposed waveform mixture algorithm, indicating the detection of leads on sea ice, which is shown in Figs. 4b and c. While the performance of the waveform mixture analysis was comparable to the DT algorithm from Lee et al. (2016), the waveform mixture analysis slightly over-estimated leads resulting in a lower user's accuracy for leads than that by DT (Figs. 4 and 5). These are inevitable results because waveforms used in the waveform mixture algorithm are basically extracted by DT from Lee et al. (2016). The lead classification results should be assessed during all the months (i.e., January to May, and October to December) and years (i.e., 2011 to 2016) using MODIS images to thoroughly evaluate the proposed waveform-based algorithm for lead detection. However, the lead classification results in January, February, November, and December were not assessed using MODIS images due to polar nights. Thus, the lead classification results in these months could possibly have uncertainties. It should be also noted that the validation was limited as the MODIS images did not fully cover the entire Arctic region (top in Fig. 4).

While all classification methods produced high producer's accuracy for the ice class exceeding 93 %, the approaches from Lee et al. (2016) and Laxon et al. (2013) resulted in a bit higher producer's accuracy for leads than the other methods

(Fig. 5). Although Laxon et al. (2013) produced the highest producer's accuracy for leads (i.e., 94.5 %), which means that this method robustly detected leads, the user's accuracy for ice was the lowest, suggesting a huge number of false alarms for leads on the ice. Overestimation of leads may increase sea surface height anomaly (SSHA), which will lead to the underestimation of sea ice freeboard (Lee et al., 2016).

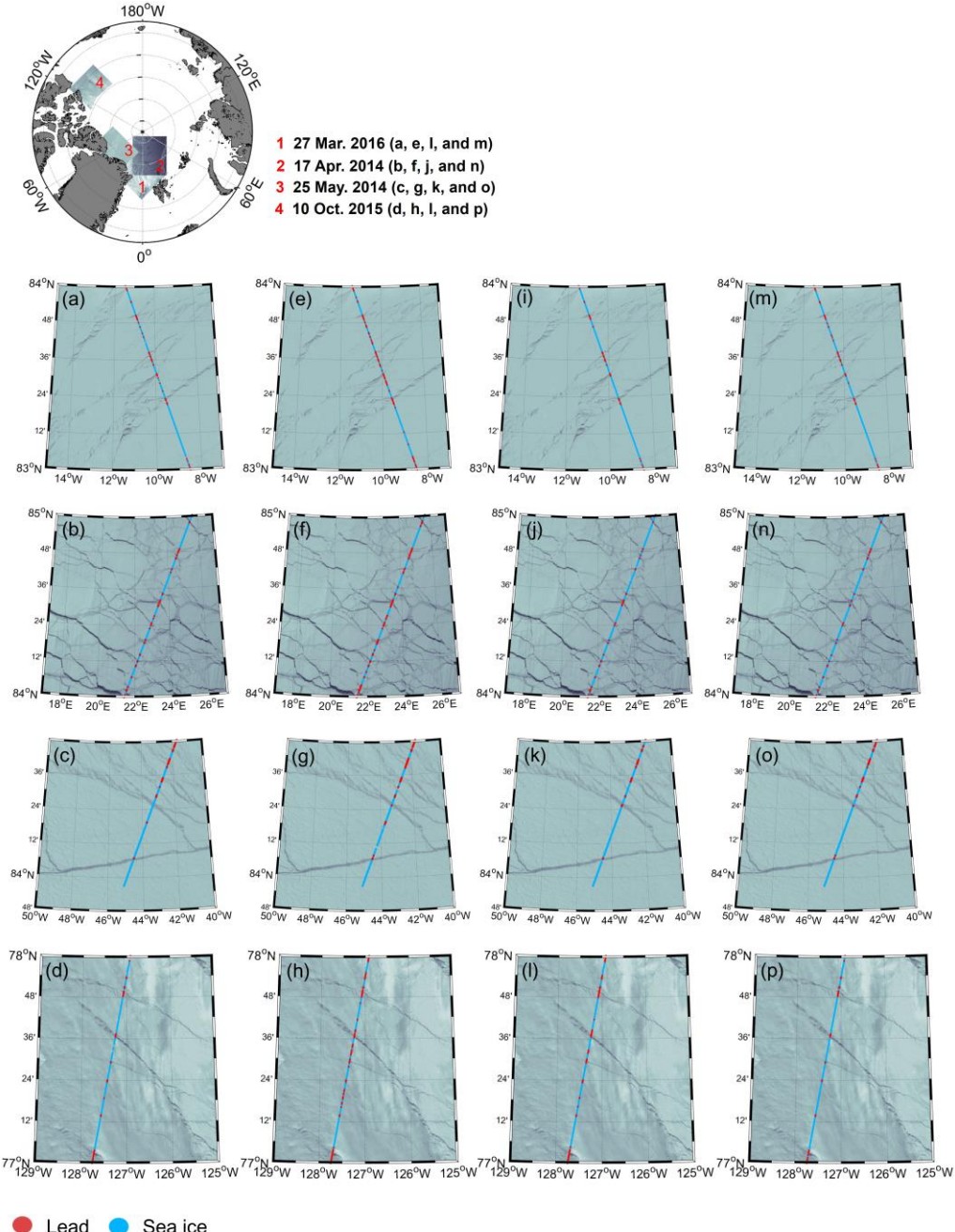

**Figure 4:** Visual comparison of lead classifications: (a) – (d) lead classifications based on Rose (2013), (e) – (h) lead classifications based on Laxon et al. (2013), (i) – (l) lead classifications based on decision trees from Lee et al. (2016), and (m) – (p) lead classifications based on the proposed waveform mixture analysis. The MODIS data were collected on 27 March 2016 (a, e, i, and m), 17 April 2014 (b, f, j, and n), 25 May 2015 (c, g, k, and o), and 10 October 2015 (d, h, l, and p). An overview map of the location of cropped MOIDS images is in top of the figure.

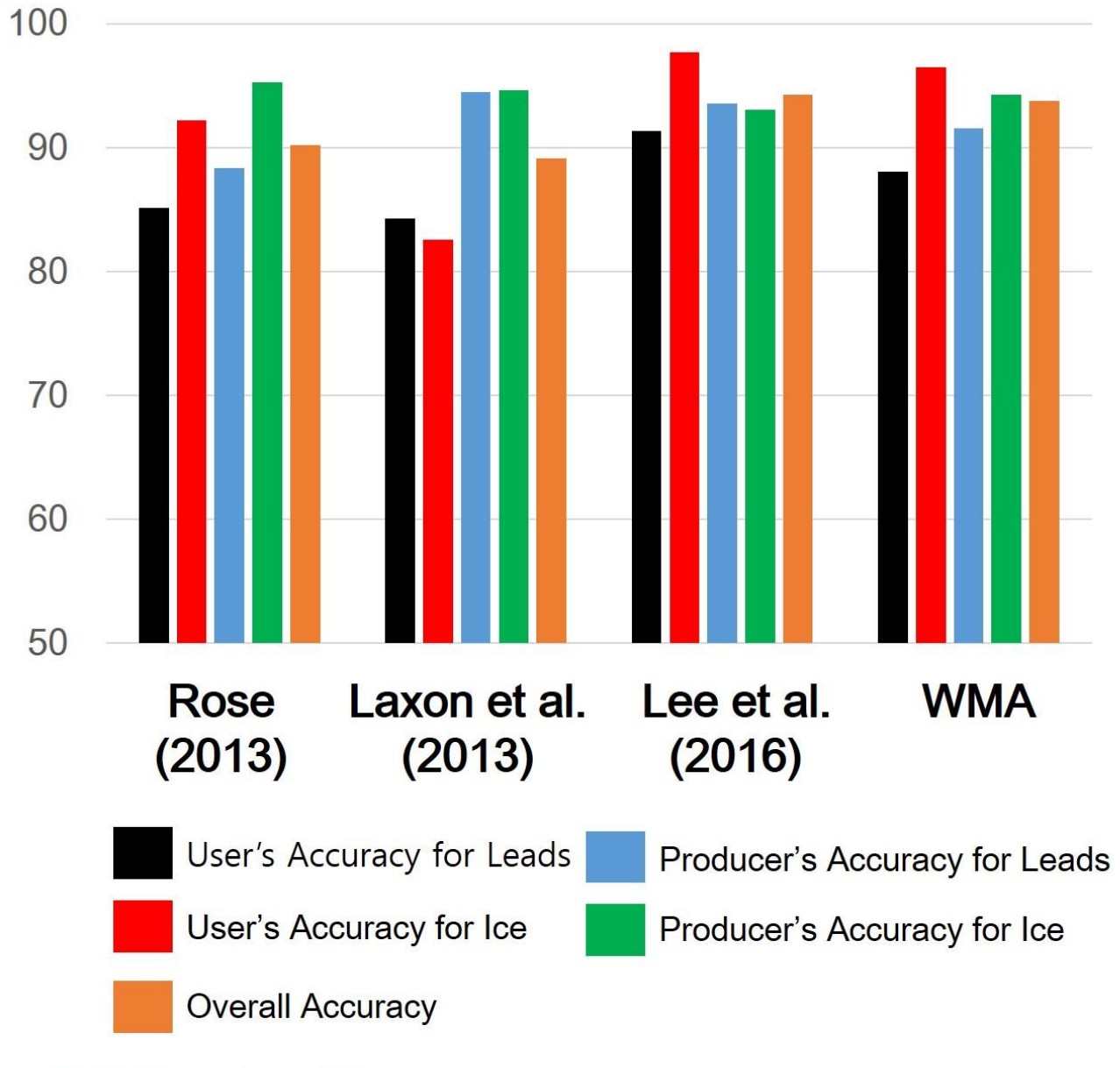

**Figure 5:** Accuracy assessment results for lead detection by method—three existing methods and the proposed waveform mixture analysis (WMA).

## 4.2 Spatiotemporal distribution of lead fraction maps

The monthly lead fraction maps with a 10 km grid in January to May, and October to December from 2011 to 2016 are shown in Figs. 6 and 7. We have compared lead fraction maps with the different spatial resolutions (i.e., 10, 50, and 100 km) to decide the proper spatial resolution. The spatial distribution of all lead fraction maps looked similar (not shown) because the ratios of lead observations to the entire CryoSat-2 observations did not significantly change among different spatial resolutions. Although the number of CryoSat-2 observations with a 10 km grid around the coastline is small (5-10), the greater number of observations in larger grids (50 and 100km) resulted in the similar distribution of lead fraction around the coastline. It is believed that the lead fraction maps with 10 km spatial resolution better represent detailed spatial distribution of leads. The areas in the marginal ice zones line of the Arctic Ocean clearly show high lead fraction due to the shear zone (i.e., an area of deformed sea ice along the coast, Serreze and Barry (2005) and outflow of sea ice. In particular, high lead fraction was found around Beaufort Sea during the spring season (MAM) because of the Beaufort Gyre, a wind-driven ocean current. It is widely known that the Chuckchi Sea is the main strait through which warm Pacific water flows into the Arctic (Woodgate et al., 2006; Woodgate et al., 2010). However, the lead fraction around the Chuckchi Sea was lower than the lead fraction around the Beaufort Sea in January to April (i.e., winter season) 2011 and 2016, excluding 2015.While the lead fraction decreases from October to March (i.e., freezing season) with the minimum in March, the lead fraction starts to increase from April. This indicates an increasing lead fraction, which corresponds to the seasonal cycle of sea ice thickness. However, the lead fraction around the Beaufort Sea decreases in March and April of 2013 and 2016 (Figs. 6 and 7). December to January is usually considered as a freezing season. Nevertheless, the lead fraction around the central Arctic increased in January 2016. This result corresponds to the findings of Kim et al. (2017) and Ricker et al. (2017).

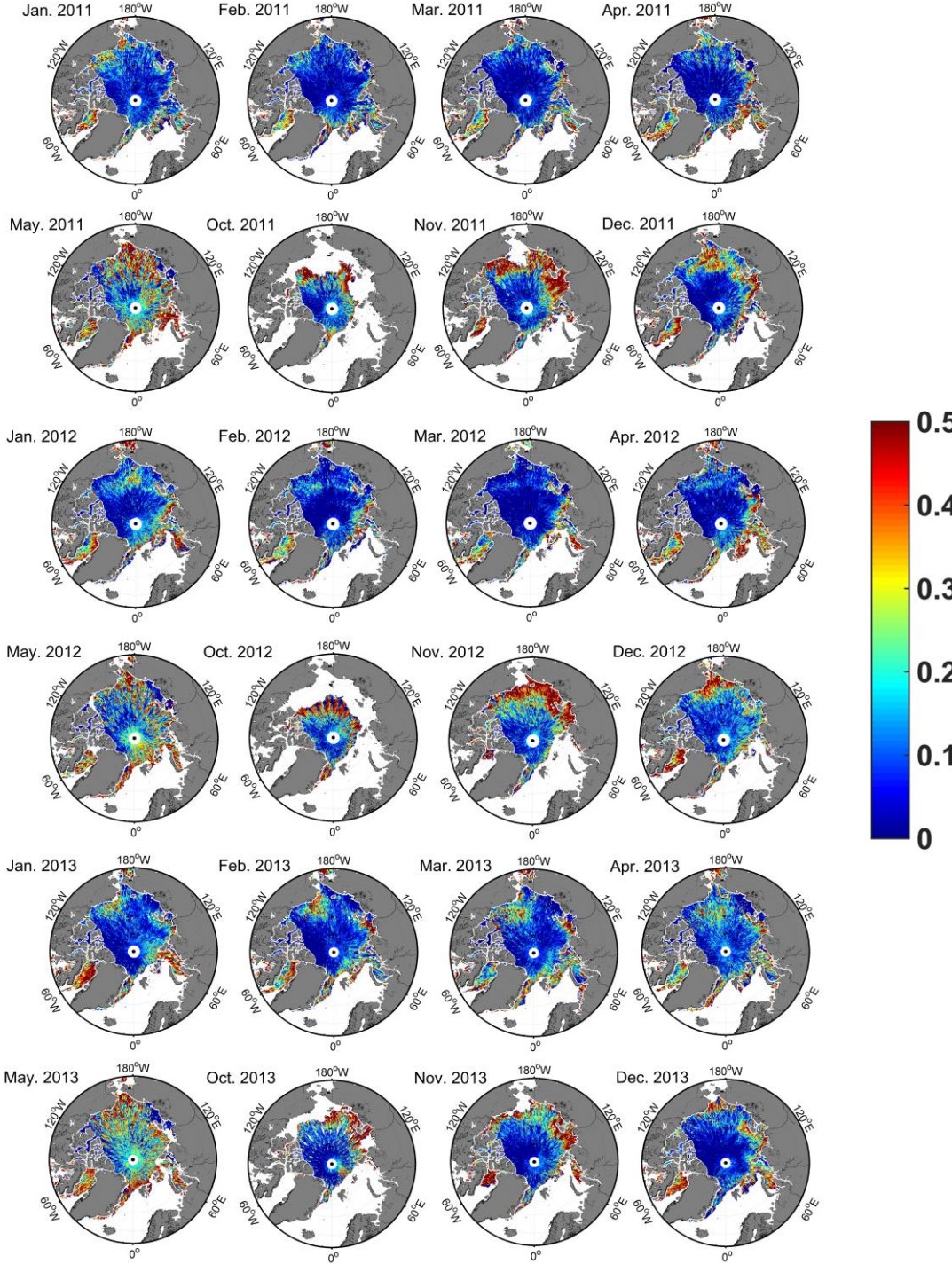

**Figure 6:** Monthly lead fraction maps based on waveform mixture analysis in January to May, October to December between 2011 and 2013. The range of the colour bar was set from 0 to 0.5 to emphasize lower values.

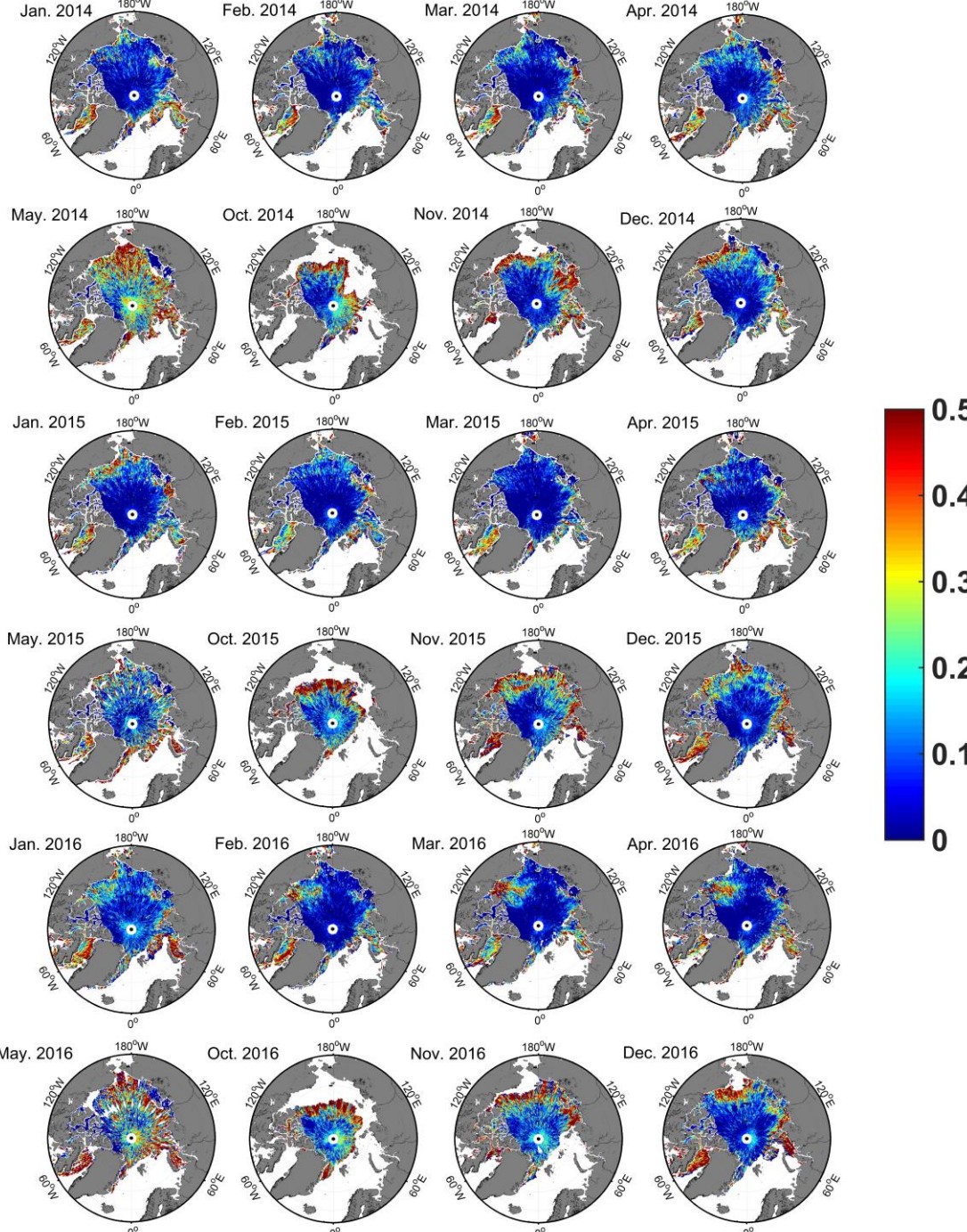

**Figure 7:** Monthly lead fraction maps based on waveform mixture analysis in January to May, October to December between 2014 and 2016. The range of the colour bar was set from 0 to 0.5 to emphasize lower values.

## 4.3 Grid sensitive analysis in 10x10 km

As mentioned in section 3.2, the number of CryoSat-2 observations decreases from the North Pole toward the coast line of Arctic Ocean. This results in an increase in statistical uncertainties when calculating monthly lead fraction around the coast line of Arctic Ocean based on the small number of CryoSat-2 observations. The number of lead and ice observations is shown in Fig. 8a-h. While there are a few lead observations in the central Arctic, a large number of ice observations was found in the central Arctic. The high standard deviation values around the coast line of the Arctic Ocean zone imply that the reliability of lead fractions was low, while the relatively large number of CryoSat-2 observations around the North Pole produced low standard deviation indicating less sensitivity (Fig. 8i-l). There was spatial difference of sensitivity by month (i.e., January to April) because of the different number of lead observations. Especially, since there was no lead observation in the East Siberian coast and Eastern Laptev Sea, the sensitivity (i.e., standard deviation) was also zero (Fig. 8c and d). It should be noted that the corresponding lead fraction might not represent actual lead fraction in a 10 x 10 km grid. This is a drawback when calculating monthly lead fraction maps with satellite altimeters.

**Number of lead observations**

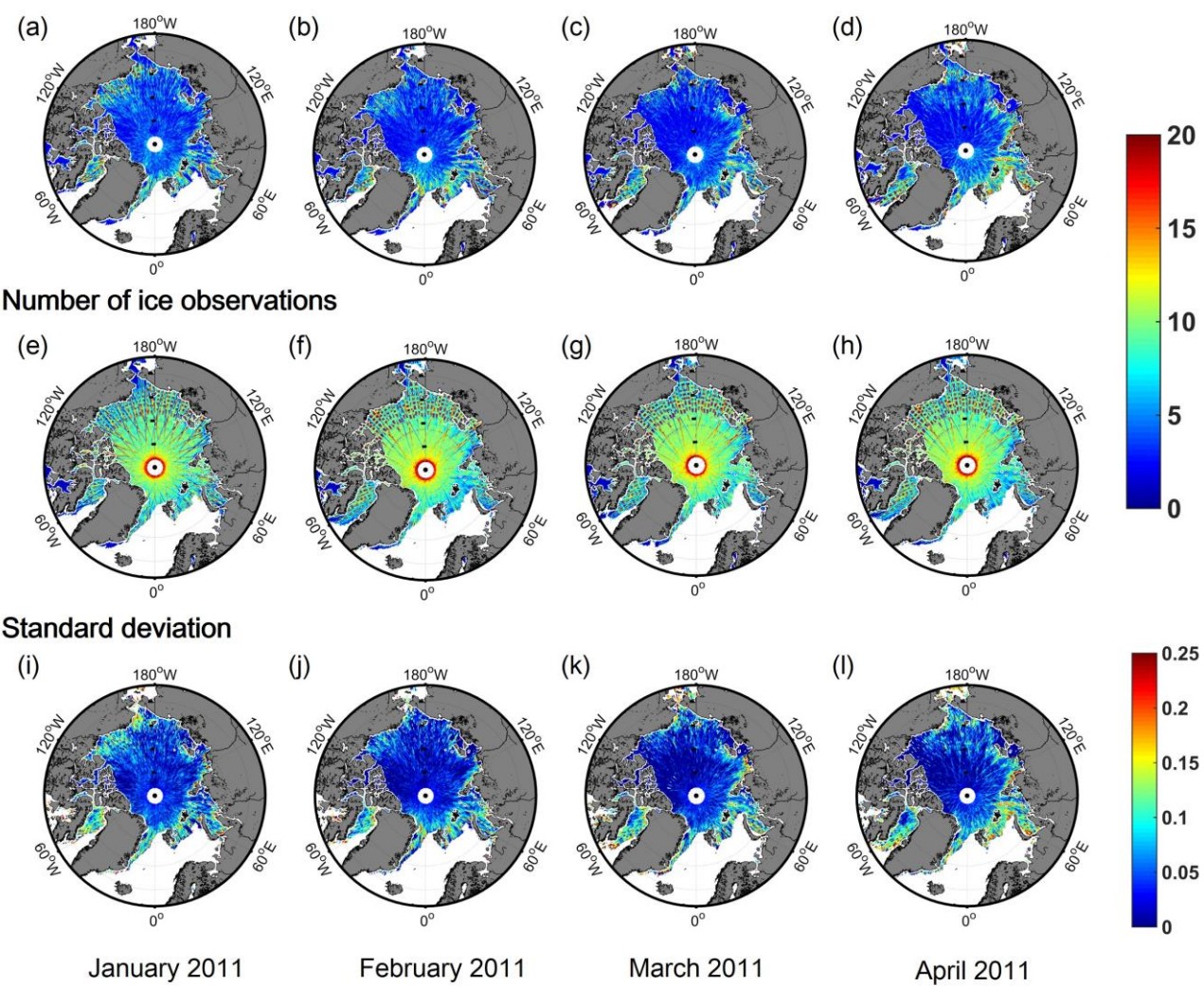

**Figure 8:** (a-d) the number of lead observations, (e-h) the number of ice observations, (i-l) the standard deviation of the results based on the sensitivity analysis of lead fraction from January to April 2011.

## 5. Discussion

### 5.1 Comparison of lead classification methods

Since the overall accuracy metrics of the proposed waveform mixture analysis approach was comparable to those of the existing methods, especially DT, the waveform-based method can be used for estimating SSHA. Threshold-based lead detection methods have to be re-scaled whenever baseline data are updated. For example, beam behaviour parameters and backscatter sigma-0 changed slightly between when baseline B and baseline C data were used. Thus, thresholds must also be updated in order to appropriately identify leads using the threshold-based methods. However, waveform mixture analysis is free from the change of baseline data because waveforms can still be used to detect leads using updated baseline data. This is the strong point of waveform mixture analysis when compared to the existing methods.

The use of waveform mixture analysis might not work well to detect leads in cases of refreezing leads. In Figs. 4 c, g, k, and o, the dark area in the MODIS scenes around the latitude of 84.26°N and longitude of 43°W was determined to be a lead class with visual inspection of the images and waveforms. Rose (2013) classified this region as ice. Laxon et al. (2013) and waveform mixture analysis detected one lead in that region. In Lee et al. (2016), DT detected more leads in that region than the other methods, but the validation could not entirely cover the dark area. In fact, since the leads are often refrozen, the shape of the waveforms in that region were likely more similar to the FYI waveform than the lead waveform (Zygmuntowska et al., 2013; Ricker et al., 2015; Lee et al., 2016). In the context of waveform mixture analysis, this region could be classified as ice. Therefore, in order to more accurately detect leads, a surface elevation anomaly is needed as well as beam behaviour parameters, backscatter sigma-0, and waveform mixture analysis because the surface elevation anomaly on refreezing leads would be low, as in other leads.

### 5.2 Comparison to other lead fraction maps

Four monthly lead fraction maps (Röhrs and Kaleschke, 2012; Wernecke and Kaleschke, 2015; Willmes and Heinemann, 2015) were compared to evaluate the pros and cons of each method used to produce the maps (Fig. 9). Basically, all four methods represent the spatiotemporal pattern of leads well for the freezing season from January to March. Scene-based lead fraction maps (i.e., AMSR-E in Figs. 9a, b and c, and MODIS in Figs. 9d, e, and f) and altimeter-based lead fraction maps (i.e., CryoSat-2 in Figs. 9g to l) have fundamentally different spatial characteristics as AMSR-E and MODIS are sensitive to different surface features. Scene-based lead fraction maps better represent the linear feature of leads and coastal polynya than altimeter-based lead fraction maps. Since the AMSR-E-based approach only detects relatively large (~ 3 km) leads, lead fractions are generally lower than in the fraction maps using the other approaches. While altimeter-based lead fractions in January 2011 (Figs. 9g and j) in the Chuckchi Sea were high, scene-based lead fractions (Figs. 9a to f) were low in January 2011. There are deformed and fragmented sea ices in the Chukchi Sea, which are different from the general lead shape. Altimeter-based lead detection methods identified leads between deformed and fragmented sea ices, generating a higher lead

fraction in the Chukchi Sea in January 2011 (Figs. 9g and j). However, scene-based lead fraction methods did not detect leads in the Chuckchi Sea well, resulting in a lower lead fraction. The MODIS-based lead detection method that used ice surface temperature (IST) did not detect leads in the Chukchi Sea (Figs. 9d, e, and f). In the AMSR-E images, sea ice signals were dominant in the footprint around the Chukchi Sea and cracks between deformed and fragmented sea ices were
identified as ice.

Altimeter-based monthly fraction maps might be insufficient to represent monthly lead fractions in the coast line of the Arctic Ocean due to the limited number of CryoSat-2 observations in a month. Nonetheless, altimeter-based lead fraction maps documented the overall spatial distribution of leads reasonably; in particular, high lead fractions in the shear zone. Wernecke and Kaleschke (2015) used a random cross validation technique to derive optimum thresholds based on ground
references (i.e., MODIS images). They identified leads conservatively to reduce false classifications. The classification results strongly depend on ground reference data. Since relatively high resolution (250m) MODIS images were used to construct reference data in this study, the waveform mixture analysis was able to identify small leads through the calibration process of the abundance data (Fig. 4). Although the proposed waveform mixture analysis produced lead fraction maps with a higher spatial resolution than those in Wernecke and Kaleschke (2015), the lead fractions around the coast line of the
Arctic Ocean from Wernecke and Kaleschke (2015) appeared to have less sensitivity. This is because of the larger number of lead observations in a much coarser grid than that from our results. The grid sensitivity analysis should be considered when interpreting the lead fraction maps around the coast line of the Arctic Ocean derived by the proposed waveform mixture analysis.

The choice of monthly lead fraction maps depends on the user's interest. Scene-based lead fraction maps better represent
coastal polynya and the intrinsic form of leads (Röhrs and Kaleschke, 2012; Willmes and Heinemann, 2016). CryoSat-2 based lead fraction maps might not represent the linear shape of typical leads well like cracks which include deformed and fragmented sea ices that are not in linear form. This is also a way to exchange heat and momentum transfer between the atmosphere and ocean, which can be detected as leads.

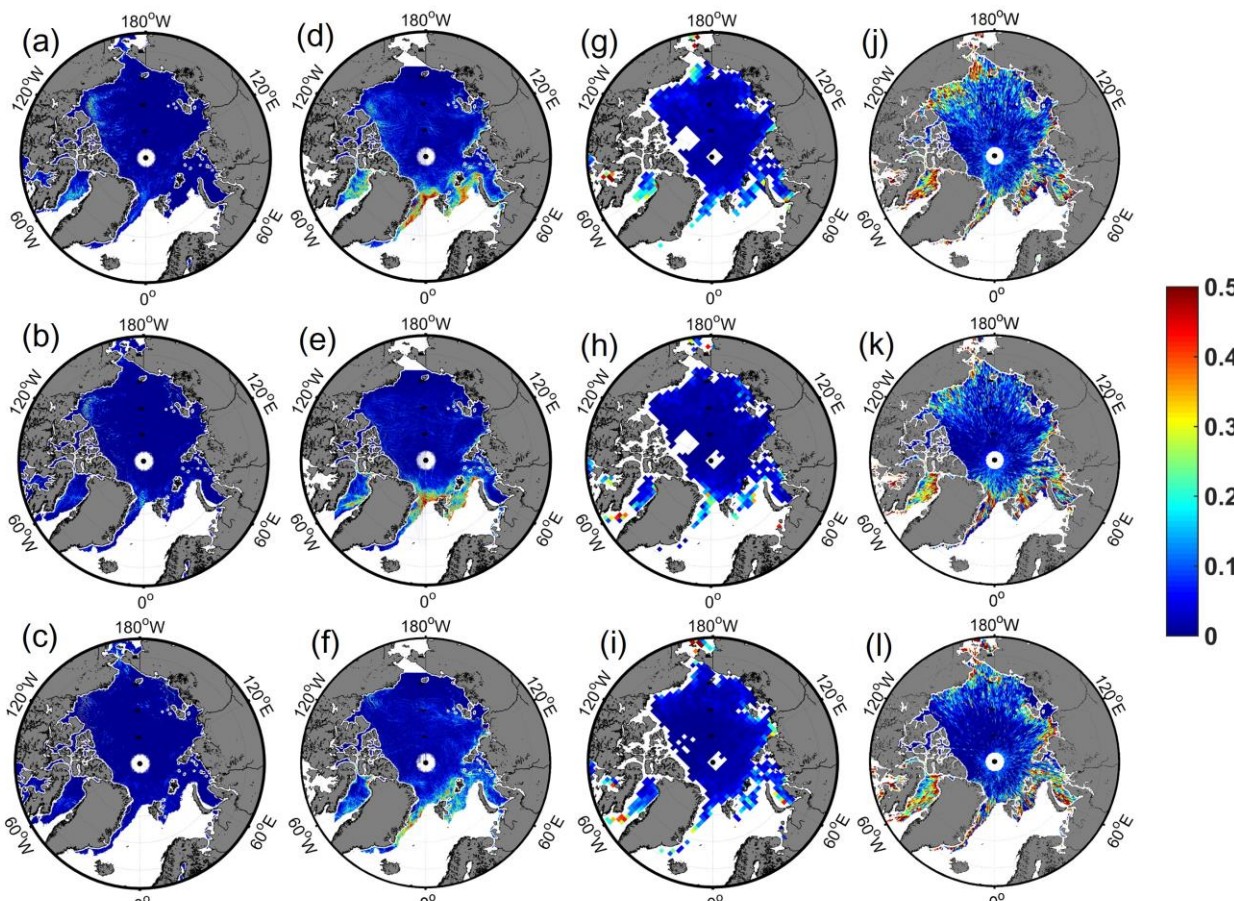

**Figure 9:** Comparison to other lead fraction maps in January to March 2011. (a-c) Monthly mean thin ice concentration maps using AMSR-E from Röhrs and Kaleschke (2011). (d-f) Monthly mean lead fraction maps using MODIS from Willmes and Heinemann (2015). (g-i) Monthly lead fraction maps using CryoSat-2 from Wernecke and Kaleschke (2015). (j-l) Monthly lead fraction maps based on waveform mixture analysis using Cryosat-2 in this study.

## 5.3 Lead dynamics

The features of the Arctic sea ice dynamics are driven by wind and, to a lesser degree, ocean currents (Kwok and Untersteiner, 2011). The Arctic Ocean circulations have contributed to the change in the state of sea ice. The lead fraction in northwestern Greenland in Fig. 9 is low because of the convergence of sea ice by two major circulations, which was clearly shown in Kwok (2015). Kwok et al. (2013) revealed that the currents speed of Beaufort Gyre and Transpolar Drift increased from the years of 1982 to 2009 and this makes the fraction of multi-year ice decrease. However, the increasing lead fraction from the years of 2011 to 2016 in this study was not seen due to the high inter-annual variability of lead fraction, particularly in the spring season (Fig. 10). High uncertainties in the marginal sea ice zone might result in not catching the increasing trend of Arctic lead fraction shown in the literature. In order to properly compare the Arctic current circulations and lead fraction, long-term lead fraction data are needed.

The inter-annual variability of lead fraction is related to atmospheric anomalous phenomena. Regarding the large inter-annual variability of lead fractions, the lead fraction in the spring season decreased from 2013 to 2014, especially around the Beaufort Sea (Figs. 6, 7, and 9). The increase and decrease of lead fractions are also linked to the change in sea ice thickness. The decrease of lead fraction in March and April from 2013 to 2014 may correspond to the increase in sea ice thickness in March and April from 2013 to 2014 (Tilling et al., 2015; Lee et al., 2016). Tilling et al. (2015) assessed the main cause of increase of sea ice thickness to be an anomalous cool summer in 2013. While November to March is considered to be the freezing season, the lead fraction increased in the central Arctic between December 2015 and January 2016 (Fig. 7). Kim et al. (2017) and Ricker et al. (2017) explained a plausible reason for the reduction in sea ice growth. Warm and moist air from the Atlantic Ocean strongly intruded into the Arctic, weakening sea ice growth. Furthermore, the high lead fraction in the Beaufort Sea in February to April 2016 was attributable to the high ice drift speed (Ricker et al., 2017).

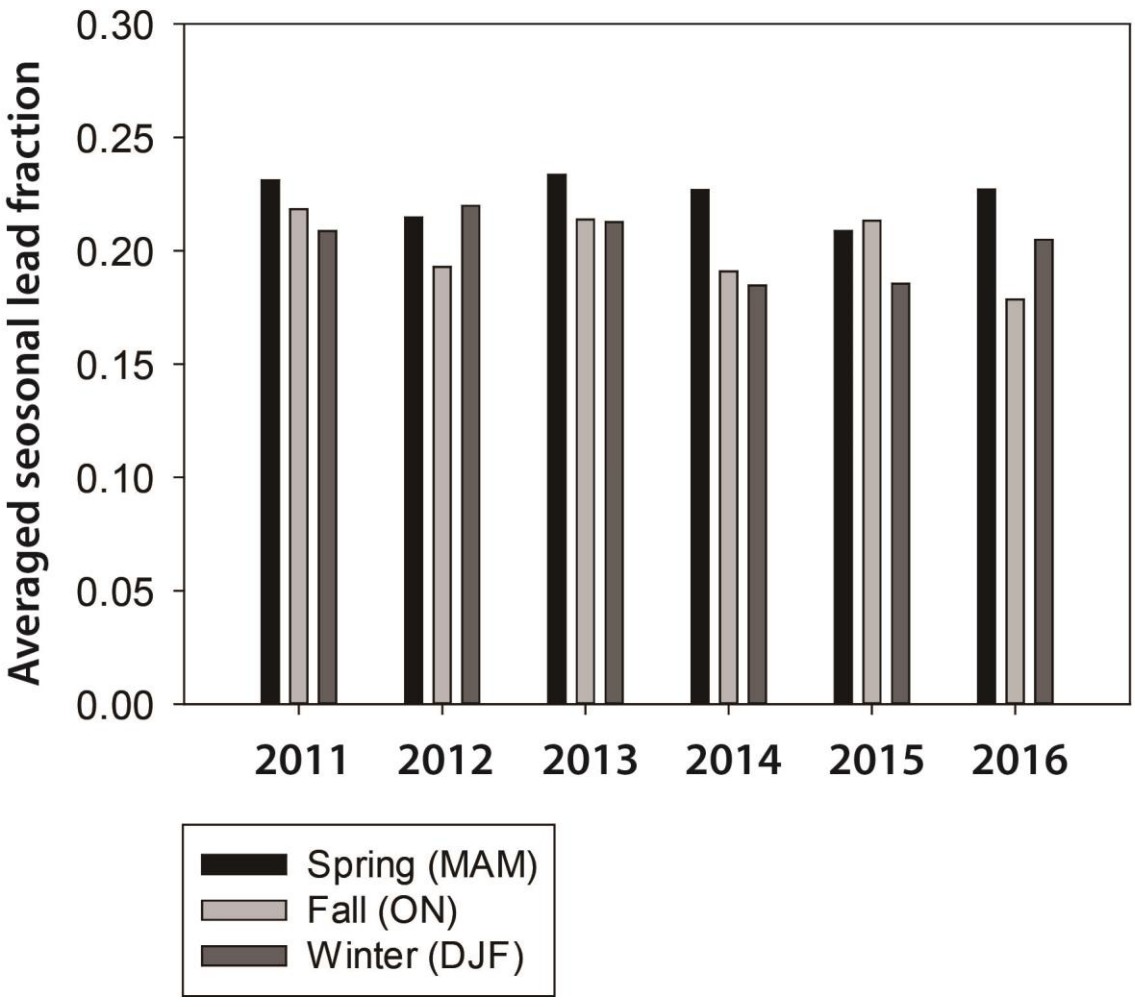

**Figure 10:** Averaged seasonal lead fraction in spring (MAM), fall (ON), and winter (DJF) between 2011 and 2016. The lead fraction from June to September was not available because leads were hard to distinguish from melt ponds using CryoSat-2 in the summer season.

## 5.4 Novelty and limitations

In this study, we developed an alternative lead detection method (i.e., waveform mixture analysis) using CryoSat-2 L1b data, which can overcome the drawbacks of previous threshold based lead detection methods. Regardless of an update in CryoSat-2 baseline data, the proposed waveform mixture analysis can consistently identify leads without rescaling parameters such as beam behaviour parameters, pulse peakiness, and backscatter sigma-0. Such parameters must be rescaled to implement threshold based lead detection methods when using updated CryoSat-2 baseline data. In addition, the proposed waveform mixture analysis outperformed the existing simple thresholding-based methods (Rose, 2013; Laxon et al., 2013), and was comparable to the machine learning-based thresholding method (Lee et al., 2016). In addition, this study showed the high inter-annual variability of Pan-Arctic lead fractions in recent years (i.e., 2011-2016), which implies that recent sea ice status has become more vulnerable to anomalous atmospheric and oceanic conditions.

On the other hand, the waveform mixture analysis depends on the quality of the endmembers. Although the use of the N-FINDR algorithm decreased the subjective selection of endmembers, waveform samples of leads and sea ice derived by DT algorithm from Lee et al. (2016) may introduce uncertainty because the algorithm was validated for March and April from 2011 to 2014. The leads that are not identifiable in the MODIS images were not considered in this study. Detecting leads smaller than the along track resolution of CryoSat-2 (~300m) with various lead detection methods should be further discussed in detail in future research using high resolution Landsat or SAR imagery. This is quite important in the retrieval of sea ice thickness using an altimeter because leads are used as the tie points for the sea surface height (SSH). For example, how the leads smaller than the along-track resolution of CryoSat-2 affect the waveform and SSH should be further investigated. The spatial resolution of monthly lead fraction maps improved up to 10km, showing a detailed spatial distribution of leads in the Arctic. For example, 10km lead fractions showed significant variations in some regions, while 50 km or 100km lead fractions did not because lead fractions are averaged, resulting in blurred spatial patterns.

## 6. Conclusions

The waveform mixture analysis was proposed to detect leads with CryoSat-2 L1b data. The lead and sea ice waveforms were considered as endmembers that are essential to implement waveform mixture analysis. The endmembers (i.e., representative waveforms of leads and sea ice) were extracted by the N-FINDR algorithm among numerous waveforms (i.e., 420,858 waveforms of sea ice and 8,501 waveforms of leads). The thresholds to make a binary classification were determined by calibrating lead and sea ice abundances with reference data extracted from high resolution (250m) MODIS images. The results show that the proposed approach robustly classified leads with comparable performance to DT from Lee et al. (2016) and slightly better than the existing simple thresholding approaches for lead detection (Rose 2013; Laxon et al., 2013). Furthermore, the lead detection of waveform mixture analysis was comparable to the decision tree based lead detection method (Lee et al., 2016), suggesting a sea ice freeboard can be retrieved with the robust lead detection method

using waveform mixture analysis. Monthly lead fraction maps were produced using the proposed waveform mixture approach, showing clear inter-annual variability. Scene-based lead fraction maps have different characteristics from altimeter-based lead fraction maps due to different sensors, algorithms, and spatial resolutions but showed similar spatial distribution. The results of the lead fraction maps are consistent with the findings of recent studies (Tilling et al., 2015; Ricker et al., 2017; Kim et al., 2017). The spatiotemporal distribution of monthly lead fraction maps were documented.

Unlike thresholds based lead detection methods, the waveform mixture analysis is less influenced on the update of baseline version of CryoSat-2 data, which will be useful for future altimeter missions. The recent strong inter-annual variability of Arctic sea ice conditions was found. In this context, this waveform mixture analysis can be used to consistently produce monthly lead fraction maps during the extended CryoSat-2 mission for monitoring Arctic sea ice.

**Acknowledgements**

This study was supported by the Korea Polar Research Institute (KOPRI) grant PE170120 (Research on analytical techniques for satellite observations of Arctic sea ice).

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
