# Peer review of "Arctic lead detection using a waveform mixture algorithm from CryoSat-2 data"

_The Cryosphere, 2017_

## Referee Comment (RC1) · S. Willmes (Referee) · 18 Sep 2017

Review of manuscript

**Sea ice leads in the Arctic Ocean: Modelling, inter-annual variability and trends**

by

Sanggyun Lee et al.

**Summary**

The paper describes a new approach for classifying Cryosat-2 waveforms along-track into leads and sea ice classes using a linear unmixing approach based on ideal endmember waveforms representative for each of the two surface types. The resulting classification is validated using MODIS reflectance data and is also compared to previously published Cryosat-2 lead retrievals. The authors find their approach to outperform existing methods and provide monthly lead fraction maps, which they compare to other available lead products. Spatial and temporal lead patterns in their results are discussed in the context of inter-annual sea-ice variability, drift and thickness.

**General comment**

While the technical approach suggested here is generally valuable and probably expands the toolbox for lead detections from Cryosat-2 data, the presentation, documentation and discussion of the results is not yet convincing and requires improvements. While the authors put forward some very strong hypotheses, some of the discussed findings are not supported by what is being shown in the figures. I think that major revisions are necessary for this study to merit publication in TC, mainly because the comprehensiveness of the paper and the degree of innovation are not yet well balanced. I suggest the paper to be worked over with emphasis being put on the validation and the documentation of the technical approach.

I will try to itemize my main critics and provide suggestions to improve the manuscript in the detailed comments below:

**Specific comments**

P2, L31: I think that point 3) is not really part of this paper. Some short discussion is provided on this topic, but not really what could be called an "investigation".

P4, L20: Here it is stated that monthly lead fraction maps were calculated. However, no further information is provided on how this was done. How did the authors deal with data gaps in the daily data? How did they convert binary data into lead fractions?

P5, L15: "The large number of ....". This statement needs some explanation.

P5, LL 21-24: What was the criterion for selecting waveforms? More information is required here.

If lead waveforms were selected using the method of Lee et al. (2016) then it should not be surprising that the results are most similar to those of Lee et al. 2016. This requires some discussion.

P5, LL25-27: What is the motivation behind comparing waveforms between months / seasons?

P6, 1st paragraph: The description of how MODIS reference data were computed needs more explanation. Please be more specific about what is compared here. The terms "examined", "assessed" and "evaluation" are used, which leaves the reader somewhat confused.

In general, the processing of the reference data (MODIS) set is not described in a sufficient way. How were MODIS swaths binarized in order to calculate the given performance metrics. Which problems might be involved?

P6, 2nd paragraph: I think this description would merit a figure to document how original waveforms end up in a binary decision on whether leads or sea ice are dominant at a given point along the altimeter track.

P6, 3rd paragraph: More explanation and documentation on the calculation of the metrics is required (equations?). How was the reference data (binary MODIS) computed?

The decision for a 10x10 km grid is not explained. Why did the authors not use a grid similar to Wernecke and Kaleschke (2015). This is especially equivocal because a) the accuracy (not sensitivity) of the lead retrieval seems to be low at lower latitudes with such a grid and b) the authors themselves state "It should be noted that the corresponding lead fraction might not represent actual lead fraction in a 10 x 10 km grid" (P11, L14).

P6, L26: What is meant by "repetitively permuted"?

P6, L27: I think it should read "…, the more sensitive the OBSERVED LEAD FRACTION IS TO THE NUMBER OF AVAILABLE OBSERVATIONS". ?

P7, L8: It is unclear to me, why 2 thresholds are necessary.

P7, LL12-14: I think that everything in parentheses can be skipped here.

P7, L20: "regardless of month": Only single swaths are presented here which does not allow a conclusion about seasonal dependencies.

P7, L21: The fact that results compare best to DT is not surprising as lead waveforms were obviously selected with the same method. This point requires to be mentioned and discussed.

P9, L9: "It is widely … ". I do not understand the subsequent argumentation with "However, the lead …"

P10, L2: It is very uncommon for Arctic surface melt to start already in April.

P10, L3: How is the lead fraction associated with the seasonal cycle of sea-ice thickness?

P10, L4: "April of 2013 and 2016" I cannot see in the figures what is being described here.

P10, Next sentence: This is a strong hypothesis. Are there references which can support it?

P10, L7: "December 2015 to .." Again, the feature described here is not in the Figure.

P10, P11, figures: The subfigures should be much larger. Please consider using one page per figure.

P10, L16: "surface elevation anomaly…" How would this help to detect leads more accurately?

P10, last paragraph: It is stated here that different spatial resolutions of the sensors are the main cause for spatial differences in monthly lead fractions. I doubt that this is the main reason. The passive microwave and thermal infrared sensors are just sensitive to different surface features as compared to the altimeter.

It is stated that there is predominantly Pancake and Nilas in the Chukchi Sea are and that this is the main cause for differences between methods. However, this is pure speculation. No support is provided for this statement. In the next paragraph, the authors even state that "Altimeter-based monthly fraction maps might be insufficient to represent monthly lead fractions in the coast line of the Arctic Ocean…", which again makes the previous discussion about the Chukchi Sea obsolete.

P12, Figure 6: It is not really clear to me, how the standard deviation is to be interpreted in terms of uncertainty. Would a higher variability of lead occurrences not also cause a higher standard deviation?

P14, L4: "well" in comparison to what?

P14, L4: What is "the shear zone"?

P14, LL10-14: Doesn't the better resolution come at a price which is too high, when the data in lower latitudes cannot be trusted?

P14, last paragraph: Again, it is stated that altimeter data are well suitable for a detection of pancake and nilas, which is by no means supported by any of the presented analysis.

P15, Figure 7: It is read to recognize details in the subfigures. Please enlarge this figure.

P15, L10: It is a bit misleading to compare Fram Strait and the Beaufort Sea in this context. The first is already part of the marginal ice zone in my mind, where lead dynamics are not primarily a result of the Transpolar Drift.

P15, L13: Six years of observations do not allow to draw conclusions about a trend.

P15, L17: "suddenly" … should be removed

P15, L18: "March and April …" The figures do not really reproduce the feature that is described here.

P15, L18 … "This unexpected decrease …. sea ice thickness " This statement needs some clarification. Why is this the case? What is actually meant here?

P16, Figure 8: If averaged data is shown and no data are available for summer, then this figure should not show continuous lines, but rather points or bars.

P16, L11: I would rather call this chapter "Discussion". Moreover, I think it could be merged with chapter 5.

P17, L4-5: "… which implies that sea ice becomes more vulnerable …". This is a strong statement which would hold only if the "recent years" could be compared to a longer time series or at least a reference.

P17, L9: How was the improvement in spatial resolution achieved? Wasn't that an arbitrary choice, which is hard to justify given the high uncertainty in the lower latitudes?

P17, L18: Please consider talking of "fractions" instead of "abundances"

P17, L19: Not clear what is meant by "regardless of month"? This is not what can be inferred from Figure 2.

Technical corrections

P1, L12: "…detect leads."   Add "from Cryosat-2 data"

P1, L20:  "…from the literature". It is rather "…from previously published data sets."

P1, L22: replace "known as" with "referred to as"

P1, L22: replace "between sea ices" with "in sea ice"

P1, L26 and throughout the manuscript: the "u" in "Lupkes" must be "ü". This holds also for "Röhrs" and "Bröhan"

P1, L27: rephrase "…could make near surface temperature up to a 3.5 K…"

P1, L29: "atmospheric boundaries". Better: "atmospheric boundary layer" ?

P2, L4,5: remove commas around "…with 1 km spatial resolution…"

P2, L10: replace "produced" with "to produce"

P2, L11: add "channels" after "(AMSR-E)"

P2, L13: Replace "could" with "can"

P2, P2, LL13-18: This paragraph needs rephrasing.

P2, L19: "…pixel ARE a linear combination…"

P2, L22: remove "the" after "However"

P2, L23: replace "enough" with "sufficient"

P2, L24: SIRAL instead of SAR ?

P2, L25: replace "…in January…" with "from January…"

P2, L26: Please add here from which data set N-FINDR selects endmembers. What is meant here by "mitigate"?

P2, L27: "evaluated" or rather "validated"?

P2, L30: replace "2) identify…" with "2) compute…"

P3, L7: "SAR and SIN". These abbreviations are not introduced. Explanation is required on what is the difference.

P3, L9: Merge parentheses after PRF

P3, L12: Remove "…and this is called multi-looking"

P3, L13: Replace "The results in the …" with "Exemplary results of possible…"

P3, L14: "…with a condition". What is meant? Please rephrase.

P4, L21: The data source should be: "These maps are available at the PANGAEA Data Publisher for Earth and Environmental Science data (http://dx.doi.org/10.1594/PANGAEA.854411)."

P4, L28: remove "an"

P5, L7: No URL is required here.

P5, LL24-25: Too many references.

P6, L3: remove "having"

---

## Referee Comment (RC2) · Anonymous Referee #2 · 23 Oct 2017

**General comments:**

This manuscript introduces a new method for Arctic lead detection using a waveform mixture analysis. The method is evaluated by comparison of its performance with other lead detection methods over MODIS imagery. Maps of lead fraction are also presented, and compared with those from other methods. Whilst the application of waveform mixture analysis to CryoSat-2 waveforms is novel and I would like to see it published, I think major revisions are needed first. My key concerns are outlined below, followed by some specific and technical comments.

Methodology

The method for applying waveform mixture analysis to CryoSat-2 waveforms is not clearly explained, and a number of assumptions are not well justified. I suggest that Section 3 requires major revision. Some specific examples:

- On P5 L1-2 the authors state that proper selection of endmembers is essential. At this point they should explain what is meant by an endmember and endmember vectors, before the linear mixture model is defined. The brief description on P5 L9, which follows model definition, is insufficient.
- P5 L21: How/why were these particular CryoSat-2 files selected?
- P5 L25-28: I do not agree that the comparison of waveforms from March and April 2011-2014 to waveforms over the much broader timeframe (in terms of months and years) of January to May, and October to December 2011-2016 is sufficient to justify the use of DT for waveforms extraction. Ideally the comparison needs to be extended to the full timeframe. Furthermore, the authors should describe how such a comparison was done.
- P6 L10: How/why were these particular images selected?
- P6 L25-26: How were the observations permuted (by what increment), and why only 30% of the observations?
- P6 L29: The authors should justify why January to April 2011 was chosen as the timeframe for comparison of lead fractions from waveform mixture analysis and existing lead detection methods. This is especially important, as the selection of a different timeframe could alter the results outlined in Section 4.3.

Evaluation

The evaluation outlined in the manuscript is inconclusive, and I believe the results are oversold. Throughout the manuscript the authors state that evaluation of lead classification with MODIS imagery has shown "better performance" than previous methods (e.g. abstract, conclusions, P7 L19). However, I strongly disagree that this is proven by the results outlined in Section 4.1, Figure 2 or Figure 3. The accuracy of the selection should only be evaluated based on lead statistics (user's and produce's accuracy for leads), and there does not appear to be a significant statistical difference between all methods. Whilst I encourage the authors to discuss the subtle similarities and differences that they have found between the accuracies of each method I do not agree with their conclusion. It is stated in the discussion Section 5.1 (P13 L4-5) that "the overall accuracy metrics of the proposed waveform mixture analysis approach was comparable to those of the existing methods", which is far more representative of the results shown.

**Specific comments:**

P1 L17: State which parameters (beam behaviour, pulse peakiness…)

P1 L23: Why only mention spring and winter? Leads are a common feature in all seasons, and the manuscript covers fall in addition to spring and winter (e.g. Figure 8). If the authors are defining seasons as certain months, then they need to be defined at this point in the text.

P1 L23 and L24: "large" and "huge" are meaningless words, without any quantification. This is an issue throughout the manuscript and I suggest the authors check for and remove all such adjectives.

P3 L7: Define SAR and SARIn

P7 L6: Are the authors basing this similarity on visual analysis only?

P7 L16: A description is needed for how the parameters were rescaled. This is crucial as the authors display these results as being representative of other methods (Rose, 2013; Laxon et al. 2013) after application to Baseline-C CryoSat-2 data, which may not be true. Related to this, it should be made clear that the 'Rose (2013)' and 'Laxon et al. (2013)' results are the authors own version, using the methodologies outlined in the related papers and therefore may differ from the actual results.

Figure 2 caption: State day and year of images, in addition to month

P10 L2: There are no letters on Figures 4 or 5

Figures 4 and 5: Whilst I appreciate the overview these figures provide, it is hard to see much detail, such as the higher lead fraction mentioned in spring 2013. There is also no logic in separating into two figures. The data would be better represented by maps for a single year to show seasonal progression of lead fraction, and a time series of mean lead fraction for each month to show inter-annual variability.

P12 L8: Again, more transparency needed that these are a reproduction of Laxon et al. (2013) results

P13 L14: From Figure 2k it appears that the Lee et al. (2016) DT method detects more than one lead in the region. This should be discussed here.

P13 L15: "typical" according to who? If some methods do detect a lead without surface elevation information how can the author be sure that a lead is not present?

P15 L13: Do not make a claim on trends, for only 6 years of data

P15 L14: The months corresponding to spring should be defined in the text, not just the figure caption. Do this at the point in the paper when seasons are first introduced.

Section 4.3. I would appreciate a comment on how these result may have been impacted by the timeframe selected by the authors. See also my concerns regarding methodology, final bullet.

**Technical comments:**

P1 L19: 2011-2016, rather than 2006-2011

P1 L22: "sea ices" to "sea ice"

P1 L23: "exchanges" to "exchange"

P1 L27: Change to "…could increase near surface temperature by 3.5 K…" or similar

P2 L4: "Recently, **the** Moderate…"

P2 L11: "…of **the** Advanced…"

P2 L13: Change to "Radar altimeters can detect leads as well"

P2 L24: "produced" to "produce"

P3 L3: "…carrying **the** Synthetic…" Check for missing "the" throughout the manuscript please. I won't correct any more.

P3 L3: "burst" to "bursts"

P3 L10: "…**so-called** Doppler beams…" (for clarity)

P11 L7: "…**statistical** uncertainties…"

P14 L1: "Figs. 8" to "Figs. 7"

---

## Referee Comment (RC3) · Anonymous Referee #3 · 14 Nov 2017

Summary

The paper proposes a novel waveform mixture analysis to detect leads, adopting the concept of linear mixture analysis that is widely used in the field of hyperspectral image analysis. The authors conclude that this method shows a better performance in detecting leads than previous methods. Moreover, spatiotemporal patterns and interannual variability of Arctic-wide lead fractions are discussed.

General Comments:

The method looks interesting, though I am not really sure if it really outperforms other existing methods. Following the given explanations is not easy and the discussion is sometimes superficial. I try summarize my major concerns:

[Figure]

1) The given results/figures do not sufficiently support the conclusions in the paper. For example, in the Conclusions section, the authors state that "The lead dynamics based on monthly lead fraction maps were examined with the Arctic Atmospheric and oceanic circulations". Where is this shown? I am also not sure if the differences between the considered algorithms are statistically significant; nor if the evaluation with MODIS images is sufficient, since the resolution is about 250 m, meaning that smaller leads detected with CryoSat-2 might be missed. See also the specific comments below.

2) The method description in the paper lacks more detailed information in the methodical part. The methods/algorithms are explained very briefly, e.g. "N-FINDER", "MATLAB toolbox for linear unmixing with the interior point least square algorithm". Although the authors refer to literature sometimes, these methods should be explained in more detail, since they are essential for understanding the study. Specifically, in the beginning, it should be explained what "endmembers" are and what they represent. The same applies to the "abundance fraction". Moreover, the authors do not show explicitly how the abundance fraction is derived. Additional figures explaining intermediate steps would be very helpful for understanding.

3) The selection of the end members needs more explanation. Where are the 48 collected CryoSat-2 orbits located? A map would be very helpful here. Also, I wonder how this approach deals with different ice types, given that first-year ice waveforms are different from multiyear ice waveforms? Reading section 3.1, it sounds like only first-year ice waveforms have been considered? The next issue is the nonlinear mixing as mentioned correctly. Due to the specular reflection, a lead (of a certain size) will always dominate the waveform. This is even more the case for the Doppler SAR processed waveforms. How is this handled in the WMA?

4) One of the objectives is to "investigate the relationship between Arctic lead fraction and thermodynamics and ice dynamics". However, this is discussed just very briefly in Section 5.3. As mentioned above, a thorough examination of the linkage to atmospheric forcing and ice dynamics is not shown.

The issues, listed above, should be addressed by the authors. Moreover, some sentences are unclear or imprecise (see specific comments). Taken together, these omissions mean that major revisions are needed.

Specific Comments:

P1L22. "sea ices" - the plural of sea ice sound odd.

P2L27: "could make near surface temperature up to a 3.5 K" - this sentence is confusing and should be rewritten. I suppose you mean that an increase in lead fraction leads to an increase in near-surface temperature of up to 3.5 K?

P3L5: "CryoSat-2 takes an advantage of SIRAL to detect smaller leads (e.g., $\sim$ 300 m)" - why should the lead size relate to the Doppler beam footprint (300 m)? The actual size of the lead might be smaller, since the specular return from the mirror-like lead surface will dominate the waveform, also if the illuminated surface is a mixture of sea ice and lead.

P3 Section 2.1: It should be clearly written which data are used here (I suppose level 1b). Which Baseline has been used (C?)? Which period is considered?

P3 Section 2.2: Same as above: Which data product version has been used? Which period?

P5L5: "vectors" -> vector

P5L21: What do you mean with "CryoSat-2 files"? An orbit file?

P5L25: "Waveforms from March to April between 2011 and 2014 were compared to those from January to May, and October to December between 2011 and 2016 (not shown), resulting in little difference between them" - Why do you separate between the two periods (January-May, October-December)?

P7L18-26: The MODIS resolution is about 250 m. What about smaller leads (< 250 m)? Due to their specular surface, they could be detected by CryoSat-2, but not with

MODIS. Therefore, I wonder how representative this evaluation is?

Figure 2: An overview map with the locations of the MODIS images would be helpful.

Figure 3: Unit is missing. Percentage?

P9L7: "The areas around the coast line" - To me it seems that lead fraction is higher at the ice edge and in the marginal ice zones, like Barents Sea?!

P11L10-12: I would argue that this conclusion is not valid: You average over one month, so leads are also propagating, opening and closing during that period. Certainly, when you have large ice drift, like in the Beaufort Gyre. Therefore, the standard deviation might not be reflect the uncertainty here.

Figure 6: Unit is missing. Percentage?

Figure 6: Why do you get these orbit patterns in the sea ice fraction maps, certainly in February 2011?

P17L3-5: "In addition, this study showed the high inter- annual variability of Pan-Arctic lead fractions in recent years (i.e., 2011-2016), which implies that sea ice becomes more vulnerable to atmospheric and oceanic forcing." - How does the interannual variability of lead fractions imply that sea ice becomes more vulnerable to atmospheric and oceanic forcing? This is not clear to me.

---

## Author Comment (AC1) · 11 Jan 2018

**Authors' responses (TC-2017-170)**

The authors would like to thank the editor and the reviewers for their precious time and invaluable comments. The corresponding changes and refinements are highlighted in yellow in the revised paper and are also summarized in our responses below. Authors' responses are in blue. Reviewer's comments are in black. When the manuscript is cited, it is shown in italics.

**Reviewer #1:**

**General comments**

While the technical approach suggested here is generally valuable and probably expands the toolbox for lead detection from CryoSat-2 data, the presentation, documentation and discussion of the results is not yet convincing and requires improvements. While the authors put forward some very strong hypotheses, some of the discussed findings are not supported by what is being shown in the figures. I think that major revisions are necessary for the study to merit publication in TC, mainly because the comprehensiveness of the paper and the degree of innovation are not yet well balanced. I suggest the paper to be worked over with emphasis being put on the validation and the documentation of the technical approach.

I will try to itemize my main critics and provides suggestions to improve the manuscript in the detailed comments below:

Thank for your comments. We agree with your major comments and significantly revised the manuscript according to your comments and those from the other reviewers. We improved insufficient technical explanation for better understanding of readers.
The major improvements of the revision are as follows:

1) The choice of spatial resolution of lead fraction maps with 10 km grid is explained.
2) Assumptions and speculations in the manuscript were removed or revised.
3) In addition, misleading statements were corrected by reviewer's comments.
4) Figures were updated and newly added with more clarity.
5) Clarity of the text was significantly improved.

The authors also would like to thank for the reviewer's minor and technical comments.

Please review our responses below and revised manuscript for details.

**Specific comments**

P2, L31: I think that point 3) is not really part of this paper. Some short discussion is provided on this topic, but not really what could be called an "investigation".

➔ Thank you for your advice. We replaced the word, "investigation" with "briefly examine" in P3 L4.

P5, L20: Here it is stated that monthly lead fraction maps were calculated. However, no further information is provided on how this was done. How did the authors deal with data gaps in the daily data? How did they convert binary data into lead fractions?

➔ We added additional explanations in terms of calculating MODIS-based monthly lead fractions in P5, L4-5.

*"The lead class was only considered to calculate daily binary lead fraction maps. The sum of the lead pixels was divided by days in a month (i.e., 28, 30, or 31) to make monthly lead fraction maps."*

P5, L15: "The large number of ….". This statement needs some explanation.

➔ We revised the sentence in P6, L5-6.

*"Since CryoSat-2 data have a large number of range bins, indicating higher vertical resolution than the range bins from Geosat, they could be used to reduce the overestimation of leads."*

P5, L21-24: What was the criterion for selecting waveforms? More information is required here.

➔ We added more information in terms of selection of waveforms in P6, L17-20.

*"The lead and sea ice endmembers (i.e., the most representative waveforms) are a key factor in the successful implementation of the waveform mixture algorithm. In order to avoid the subjective selection of endmembers, a number of endmember candidates were extracted by the DT algorithm (Lee et al., 2016) and the N-FINDR algorithm determined the optimum lead and ice endmembers."*

P5, L25-27: What is the motivation behind comparing waveforms between months/seasons?

➔ The DT model from Lee et al. (2016) was developed using data (i.e., stack standard deviation, stack skewness, stack kurtosis, pulse peakiness, and backscatter sigma-0) collected in March and April 2011-2014. Thus, the waveforms in other months and years should be compared to the waveforms in March and April 2011-2014 to examine whether the waveforms derived by the DT model during the study period are appropriate to implement the waveform mixture analysis. We described it in P7 L5-8.

*"The DT model from Lee et al. (2016) was developed using data (i.e., stack standard deviation, stack skewness, stack kurtosis, pulse peakiness, and backscatter sigma-0) collected in March and April 2011-2014. Thus, the waveforms in other months and years should be compared with the waveforms in March and April 2011-2014 to identify whether the waveforms derived by the DT model during the study period are appropriate to implement the waveform mixture algorithm."*

P6, 1st paragraph: The description of how MODIS reference data were computed needs more

explanation. Pleased be more specific about what is compared. The terms "examined", "assessed", and "evaluation" are used, which leaves the reader somewhat confused.

→ We used Earth View 250m Reflective Solar Bands Scaled Integers in MOD02QKM and just adjusted the contrast to emphasize leads from sea ice in the images. We did not any calculation with MODIS data. We added additional explanation in P7, L14-15.
*"We used Earth View 250m Reflective Solar Bands Scaled Integers in MOD02QKM and adjusted the contrast to emphasize leads from sea ice in the images."*
We unify the use of the word to evaluate and evaluation in the paragraph in P7 L13.

P6, 2nd paragraph: I think this description would merit a figure to document how original waveforms end up in a binary decision on whether leads or sea ice are dominant at a given point along the altimeter track.

→ We added Figure 3, which contains lead and ice abundances with an explanation to better describe the waveform mixture algorithm in P9 (Response Fig. 1).

[Figure]

**Response Fig. 1** Lead and ice abundance derived by waveform mixture analysis on 10 Oct. 2015. (a) Lead abundance, (b) Ice abundance. The colour bar expresses abundances from 0 to 1.

P6, 3rd paragraph: More explanation and documentation on the calculation of the metrics in required (equations?). How was the reference data (binary MODIS) computed?

→ We added a table (an error matrix to calculate user's and producer's accuracies, and overall accuracy for lead/ice classification) and the equations of each accuracy metric in P10, L1-7 (Response Tab. 1). The calculation of accuracy metrics is explained in P9 L4-10.
*"Lead detection results were evaluated using three accuracy metrics—producer's accuracy, user's accuracy, and overall accuracy (Tab. 1). Producer's accuracy (i.e., $a/(a+c)$ in the table), which is associated with omission errors, is calculated as the*

*percentage of correctly classified pixels in terms of all reference samples for each class. User's accuracy (i.e., a/(a+b) in the table), which is related to commission errors, is calculated as the fraction of correctly classified pixels with regards to the pixels classified to a class. Overall accuracy (i.e., (a+d)/(a+b+c+d) in the table) is calculated as the total number of correctly classified samples divided by the total number of validation sample data."*

The lead and ice reference data using MODIS images and CryoSat-2 tracks were labelled through visual interpretation. It is documented in P9 L9-10.

*"The lead and ice reference data using MODIS images and CryoSat-2 tracks were labelled through visual interpretation."*

**Response Tab. 1** Error matrix for calculation of user's, producer's and overall accuracy in terms of lead and ice classification.

|  |  | MODIS references | | |
|---|---|---|---|---|
|  |  | **Lead** | **Ice** | **Sum** |
|  | **Lead** | a | b | (a+b) |
| CryoSat-2 based classification | **Ice** | c | d | (c+d) |
|  | **Sum** | (a+c) | (b+d) | (a+b+c+d) |

P6, 4[th] paragraph: The decision for a 10x10 km grid is not explained. Why did the authors not use a grid similar to Wernecke and Kaleschke (2015). This is especially equivocal because a) the accuracy (not sensitivity) of the lead retrieval seems to be low at lower latitudes with such a grid and b) the authors themselves state "It should be noted that the corresponding lead fraction might not represent actual lead fraction in a 10x10 km grid" (P11, L14)

➔ We compared lead fraction maps with the different spatial resolutions (i.e., 10, 50, and 100 km) to decide the proper spatial resolution. The spatial distribution of all lead fraction maps appeared similar (Response Fig. 2 below) because the ratios of lead observations to the entire CryoSat-2 observations did not significantly change among different spatial resolutions. Although the number of CryoSat-2 observations with a 10 km grid around the coastline is small (5-10), the greater number of observations in larger grids (50 and 100km) resulted in the similar distribution of lead fraction around the coastline. Thus, we think that the lead fraction maps with 10 km spatial resolution better represent detailed spatial distribution of leads. We have further tested how the

standard deviation (i.e., sensitivity) changes according to different spatial resolution (i.e., 10 km and 50 km) (Response Fig. 3).

➔ We added explanation in P15 L2-9.

*"We have compared lead fraction maps with the different spatial resolutions (i.e., 10, 50, and 100 km) to decide the proper spatial resolution. The spatial distribution of all lead fraction maps looked similar (not shown) because the ratio to lead observations and to CryoSat-2 observations is not significantly changed among different spatial resolutions. Although the number of CryoSat-2 observations with a 10 km grid around the coastline is small (5-10), the greater number of observations in larger grids (50 and 100km) resulted in the similar distribution of lead fraction around the coastline. It is believed that the lead fraction maps with 10 km spatial resolution better represent detailed spatial distribution of leads."*

[Figure]

**Response Fig. 2** The different spatial resolution of lead fraction map (a) 10 km, (b) 50 km, and (c) 100 km.

[Figure]

**Response Fig. 3** Sensitivity analysis in terms of grid size between 10 km and 50 km. Left Column is 10 km spatial resolution below 75°**N**. Right column is 50 km spatial resolution below 75°**N**

➔ Actual lead fraction in this manuscript (P18, L11) means that lead fraction (ratio lead area to ice area) within a specific area (i.e., 1 km) like a MODIS image. However, lead fraction in this study is just ratio of lead observations to the given CryoSat-2 observations within a 10 km grid because we use the altimeter to calculate monthly lead fraction.

P6, L26: What is meant by "repetitively permuted"?

P6, L27: I think it should read "…, the more sensitive the OBSERVED LEAD FRACTION IS TO THE NUMBER OF AVAILABLE OBSERVATIONS".

➔ We revised the sentence. We wanted to say that the number of lead and ice observations in a grid randomly changed for each iteration.

We added above explanation in P10, L10-13.

*"Thirty (30) percent of the lead and ice observations in 10x10 km grids was randomly permuted 50 times, and the standard deviation of the resultant lead fractions through the 50 iterations were calculated by grid. The higher the standard deviation in a grid, the more sensitive the observed lead fraction is to the number of available observations."*

P7, L8: It is unclear to me, why 2 thresholds are necessary.

➔ Since the waveform mixture algorithm produces two abundances (i.e., lead and ice abundances), the optimum thresholds have to be determined to distinguish leads. This becomes clearer as Fig. 3 is added in the manuscript. The basic rule is "if the lead abundance of a pixel is greater than X and its ice abundance is smaller than Y, then assign LEAD to the pixel."

P7, L12-14: I think that everything in parentheses can be skipped here.

➔ Thank you for your comment. We removed the parentheses.

P7, L20: "regardless of month": Only single swaths are presented here which does not allow a conclusion about seasonal dependencies.

➔ Thank you for your comment. We removed the phrase.

P7, L21: the fact that results compare best to DT is not surprising as lead waveforms were obviously selected with the same method. This point required to be mentioned and discussed.

➔ We agree with your opinion and added a sentence in P11, L24-25.

*"These are inevitable results because waveforms used in the waveform mixture algorithm are basically extracted by DT from Lee et al. (2016)."*

P9, L9: "It is widely…". I do not understand the subsequent argumentation with "However, the lead…".

➔ We revised the sentences in P15, L12-14.

*"It is widely known that the Chuckchi Sea is the main strait through which warm Pacific water flows into the Arctic (Woodgate et al., 2006; Woodgate et al., 2010). However, the lead fraction around the Chuckchi Sea was lower than the lead fraction around the Beaufort Sea in January to April (i.e., winter season) 2011 and 2016, excluding 2015."*

P10, L2: It is very uncommon for Arctic surface melt to start already in April.
➜ Thank you for your comment. We corrected the sentence in P15, L15
   *"The lead fraction starts to increase from April."*

P10, L3: How is the lead fraction associated with the seasonal cycle of sea ice thickness?
➜ In fact, there is no reference to this point but we checked the spatial distribution of sea ice thickness Jan. to Mar. and Nov. to Dec. using the retrieval of sea ice thickness algorithm from Lee et al. (2016). The sea ice thickness starts to increase from October to April and decreases from April. In the perspective of the spatial distribution of sea ice thickness, sea ice thickness is high for areas with low lead fraction, vice versa.

P10, L4: "April of 2013 and 2016" I cannot see in the figures what is being described here.
➜ Thanks. We removed the sentence.

P10, L5: This is a strong hypothesis. Are there references which can support it?
➜ Thanks. We think it would be better to remove the sentence.

P10, L7: "December 2015 to .." Again, the feature described here is not in the Figure.
➜ We revised the sentence in P15, L18.
   *"Nevertheless, the lead fraction around the central Arctic increased in January 2016"*

P10, P11, figures: The subfigures should be much larger please consider using one page per figure.
➜ We enlarged Figures 6 and 7 in P16-17.

P13, L16, figures: "surface elevation anomaly…" How would this help to detect leads more accurately?
➜ Refreezing leads might tend to be classified as ice using the waveform mixture algorithm (Fig. 4o), which are apparently leads with a lower elevation than ice. The surface elevation anomaly is needed as well as beam behaviour parameters, backscatter sigma-0, and waveform mixture analysis because the surface elevation anomaly on refreezing leads would be low, as in other leads.
We added above explanation in P20, L16-18.
   *"Therefore, in order to more accurately detect leads, a surface elevation anomaly is needed as well as beam behaviour parameters, backscatter sigma-0, and waveform mixture analysis because the surface elevation anomaly on refreezing leads would be low, as in other leads."*

P13, last paragraph: It is stated here that different spatial resolutions of the sensors are the main cause for spatial differences in monthly leads fractions. I doubt that this is the main reason. The passive microwave and thermal infrared sensors are just sensitive to different surface features

as compared to the altimeter.

➔ Thank you for your comment. We removed the sentence and added an explanation in P20, L23-26.

*"Scene-based lead fraction maps (i.e., AMSR-E in Figs. 9a, b and c, and MODIS in Figs. 9d, e, and f) and altimeter-based lead fraction maps (i.e., CryoSat-2 in Figs. 9g to l) have fundamentally different spatial characteristics as AMSR-E and MODIS are sensitive to different surface features."*

It is stated that there is predominantly Pancake and Nilas in the Chukchi Sea and that this is the main cause for differences between methods. However, this is pure speculation. No support is provided for this statement. In the next paragraph, the authors even state that "Altimeter-based monthly fraction maps might be insufficient to represent monthly lead fractions in the coast line of the Arctic Ocean…" which again makes the previous discussion about the Chuck Sea obsolete.

➔ Thanks for the catch. We misinterpreted sea ice in Chuck Sea. We think the Pancake and Nilas sea ice mentioned in the manuscript should be deformed and fragmented sea ices (Response Fig. 4). We try to find sea ice images in Jan. 2011 as mentioned but the image could not be found due to polar night. Response Fig. 3 is the best sea ice image collected close to Jan. 2011 without cloud interference. We changed the words from Pancake and Nilas to deformed and fragmented in P20, L30-31 and P21 L1.

*"There are deformed and fragmented sea ices in the Chukchi Sea, which are different from the general lead shape. Altimeter-based lead detection methods identified leads between deformed and fragmented sea ices, generating a higher lead fraction in the Chukchi Sea in January 2011 (Figs. 9g and j)"*

[Figure]

**Response Fig. 4** Deformed and fragmented sea ice images near Chukchi Sea from MODIS on collected on 2 Feb. 2011.

P12, Figure 6: It is not really clear to me, how standard deviation is to be interpreted in terms of uncertainty. Would a higher variability of lead occurrences not also cause a higher standard deviation?

➔ The monthly lead fraction is calculated by the ratio of the number of lead observations to the number of CryoSat-2 observations (i.e., the number of lead observations / the number of CryoSat-2 observations) within a 10 km grid. Since the number of CryoSat-2 observation around the coastline is small (5-10), the thirty percent of the number of lead and ice observation is randomly changed, resulting in a large variation of lead fraction in a grid. Meanwhile, the number of CryoSat-2 observations in higher latitudes is large (> 30). The random change of the thirty percent of the number of lead and ice observation results in a small variation of the lead fraction in a grid. The sensitivities in terms of the calculation of monthly lead fraction maps depends on the number of CryoSat-2 observations.

We replaced "uncertainty" with "sensitivity" throughout the manuscript and added above explanation in section 3.3 and 4.3

P14, L5: "Well" in comparison to what?

➔ We revised the sentence in P21, L7-8.

*"Altimeter-based lead fraction maps reasonably documented the overall spatial distribution of leads, in particular, high lead fractions in the shear zone"*

P14, L5: What is "the shear zone"?
→ The shear zone is an area of deformed sea ice along the coast (Serreze and Barry, 2005). We added this explanation in P15, L9-10.

P14, L10-14: Doesn't better resolution come at a price which is too high, when the data in lower latitudes cannot be trusted?
→ We have compared lead fraction maps with the different spatial resolutions (i.e., 10, 50, and 100 km) to decide the proper spatial resolution. The spatial distribution of all lead fraction maps appeared similar (Response Fig. 2) because the ratios of lead observations to CryoSat-2 observations did not significantly change among different spatial resolutions. Although the number of CryoSat-2 observations with a 10 km grid around the coastline is small (5-10), the greater number of observations in larger grids (50 and 100km) resulted in the similar distribution of lead fraction around the coastline. Thus, we think that the lead fraction maps with 10 km spatial resolution better represent detailed spatial distribution of leads. We have further tested how the standard deviation (i.e., sensitivity) changes according to different spatial resolution (i.e., 10 km and 50 km) (Response Fig. 3).

→ We added explanation in P15 L2-9.
*"We have compared lead fraction maps with the different spatial resolutions (i.e., 10, 50, and 100 km) to decide the proper spatial resolution. The spatial distribution of all lead fraction maps looked similar (not shown) because the ratio to lead observations and to CryoSat-2 observations is not significantly changed among different spatial resolutions. Although the number of CryoSat-2 observations with a 10 km grid around the coastline is small (5-10), the greater number of observations in larger grids (50 and 100km) resulted in the similar distribution of lead fraction around the coastline. It is believed that the lead fraction maps with 10 km spatial resolution better represent detailed spatial distribution of leads."*

P14, last paragraph: Again, it is stated that altimeter data well suitable for a detection of pancake and nilas, which is by no means supported by any of the presented analysis.
→ We changed the words from pancake and nilas to deformed and fragmented in P20, L30-31 and P21 L1.
*"There are deformed and fragmented sea ices in the Chukchi Sea, which are different from the general lead shape. Altimeter-based lead detection methods identified leads between deformed and fragmented sea ices, generating a higher lead fraction in the Chukchi Sea in January 2011 (Figs. 9g and j)."*

P15, Figure 7: It is read to recognize details in the subfigures. Please enlarge this figure.
➔ We enlarged Figure 9 in P22.

P15, L10: It is a bit misleading to compare Fram Strait and the Beaufort Sea in this context. The first is already part of the marginal ice zone in my mind, where lead dynamics are not primarily a result of the Transpolar Drift.
➔ Thank you for your comment. We removed the sentence.

P15, L13: Six years of observations do not allow to draw conclusions about a trend.
➔ Thank you for your comment. We removed the word, "trend".

P15, L17: "suddenly" … should be removed.
➔ We removed the word as suggested.

P15, L18: "March and April …" The figures do not really reproduce the feature that is described here.
➔ We revised the sentence in P23, L11-13.
*"Regarding the large inter-annual variability of lead fractions, the lead fraction in the spring season decreased from 2013 to 2014, especially around the Beaufort Sea (Figs. 6, 7, and 9)."*

P15, L18: … "This unexpected decrease …. sea ice thickness" This statement needs some clarification. Why is this the case? What is actually meant here?
➔ We wanted to say that the increase and decrease of lead fractions are linked to the change in sea ice thickness. We added more explanation in the manuscript in P23, L13-15.
*"The increase and decrease of lead fractions are also linked to the change in sea ice thickness. The decrease of lead fraction in March and April from 2013 to 2014 may correspond to the increase in sea ice thickness in March and April from 2013 to 2014 (Tilling et al., 2015; Lee et al., 2016)."*

P16, Figure 8: If averaged data is shown and no data are available for summer, then this figure should not show continuous lines, but rather points or bars.
➔ Thank you for your comment. We changed Figure 10 in P24.

P16, L11: I would rather call this chapter "Discussion". Moreover, I think it could be merged with Chapter 5.
➔ Thank you for your comment. We moved Chapter 6 "Novelty and Limitations" as a sub-section of Chapter 5 Discussion.

P17, L4-5: "… which implies that sea ice becomes more vulnerable…" This is a strong statement which would hold only if the "recent years" could be compared to a longer time series or at least a reference.

➜ We revised the sentence in P25, L8-10.

*"This study showed the high inter-annual variability of Pan-Arctic lead fractions in recent years (i.e., 2011-2016), which implies that recent sea ice status has become more vulnerable to anomalous atmospheric and oceanic conditions."*

P17, L9: How was the improvement in spatial resolution achieved? Wasn't that an arbitrary choice, which is hard to justify given the high uncertainty in the lower latitudes?

➜ We compared lead fraction maps with the different spatial resolutions (i.e., 10, 50, and 100 km) to decide a proper spatial resolution. The spatial distribution of all lead fraction maps appeared similar (Response Fig. 2) because the ratios of lead observations to CryoSat-2 observations did not significantly change among different spatial resolutions. Although the number of CryoSat-2 observations with a 10 km grid around the coastline is small (5-10), the greater number of observations in larger grids (50 and 100km) resulted in the similar distribution of lead fraction around the coastline. Thus, we think that the lead fraction maps with 10 km spatial resolution better represent detailed spatial distribution of leads. We have further tested how the standard deviation (i.e., sensitivity) changes according to different spatial resolution (i.e., 10 km and 50 km) (Response Fig. 3).

➜ The improvement of spatial resolutions is documented in P25 L19-21.

*"The spatial resolution of monthly lead fraction maps improved up to 10km, showing a detailed spatial distribution of leads in the Arctic. For example, 10km lead fractions showed significant variations in some regions, while 50 km or 100km lead fractions did not because lead fractions are averaged, resulting in blurred spatial patterns"*

P17, L18: Please consider talking of "fractions" instead of "abundances"

➜ We think the use of the word, "abundance" should be maintained as it is a common term when spectral mixture analysis is used. While the lead and sea ice abundances were produced by the waveform mixture algorithm, the monthly lead fractions were made by calculating a lead fraction in a grid based on the ratio of lead observations to CryoSat-2 observations.

P17, L19: Not clear what is meant by "regardless of month"? This is not what can be inferred from Figure 2.

➜ Thank you for your comment. We removed the words.

Technical corrections

P1, L12: "…detect leads." Add "from CryoSat-2 data"
➔ Thank you for your correction. We added "from CryoSat-2data".

P1, L20: "…from the literature." It is rather "…from previously published data sets"
➔ We corrected the sentence as suggested.

P1, L22: replace " known as" with "referred to as"
➔ We replaced the word "known as" with "referred to as".

P1, L22: replace " between sea ices" with "in sea ice"
➔ We replaced the word "between sea ices" with "in sea ice" as suggested.

P1, L26 and throughout the manuscript: the "u" in "Lupkes" must be "ü". This holds also for "Röhrs" and "Bröhan"
➔ Thank you for your correction. We added the Umlaut throughout the manuscript.

P1, L27: rephrase "…could make near surface temperature up to a 3.5 K…"
➔ We revised the sentence as suggested.

P1, L29: "atmospheric boundaries" Better: "atmospheric boundary layer" ?
➔ We revised the sentence as suggested.

P2, L4, 5: remove commas around "…with 1 km spatial resolution…"
➔ We removed the commas.

P2, L10: Replace "produced" with "to produce"
➔ We replaced "produced" with "to produce".

P2, L11: add "channels" after "(AMSR-E)"
➔ We added channels after AMSR-E.

P2, L13: Replace "could" with "can"
➔ We replaced "could" with "can".

P2, L13-18: This paragraph needs rephrasing.
➔ We rephrased the paragraph in P2, L16-22.
*"Airborne and spaceborne radar altimeters can detect leads as well. Zygmuntowska et al. (2013) used Airborne Synthetic Aperture and Interferometric Radar Altimeter System (ASIRAS), similar to CryoSat-2, to identify leads based on waveform characteristics and a Bayesian classifier. Zakharova et al. (2015) and Wernecke and*

> *Kaleschke (2015) used the spaceborne altimeters Satellite with Argos and Altika (SARAL) and CryoSat-2 to identify leads, respectively. While Zakharova et al. (2015) applied simple thresholds to identify leads along with Satellite with Argos and Altika (SARAL/Altika) tracks and estimated regional lead fractions, Wernecke and Kaleschke (2015) optimized thresholds to detect leads and produced pan-Arctic lead fraction maps using CryoSat-2 with an analysis of lead width, and sea surface height."*

P2, L19: "…pixel Are a linear combination…"
> ➜ Corrected.

P2, L22: remove "the" after "However"
> ➜ Corrected.

P2, L24: SIRAL instead of SAR?
> ➜ Thank you for your correction. We replaced "SAR" with "SIRAL."

P2, L25: replace "…in January…" with "from January…"
> ➜ We replaced "in January" with "from January".

P2, L26: Please add here from which data set N-FINDR selects endmembers. What is meant by "mitigate"?
> ➜ We rephrased the sentence in P2, L30-32.
> Over hundred thousands of waveforms were extracted by the DT algorithm. *Waveform endmembers* are crucial to implement spectral mixture analysis (Fig. 1). The N-FINDR (N-finder) algorithm was used to select representative waveform endmembers from the waveforms extracted by decision trees (DT) from Lee et al. (2016), which avoids the subjective selection of endmembers.
> *"Waveform endmembers are crucial to implement spectral mixture analysis (Fig. 1). The N-FINDR (N-finder) algorithm was used to select waveform endmembers from extracted waveforms by Decision tree (DT) from Lee et al. (2016), which avoids the subjective selection of endmembers"*

P2, L27: "evaluated" or rather "validated"?
> ➜ Thank you for your advice. We replaced "evaluated" with "validated" in P2 L32.
> *"The detected leads were visually evaluated with MODIS images (at 250 m resolution) and compared with other thresholds based lead detection methods."*

P2, L30: replace "2) identify…" with "2) compute…"
> ➜ Thank you for your correction. We replaced "identify" with "compute" in P3 L3.

**Authors' responses (TC-2017-170)**

P3, L7: "SAR and SIN". These abbreviations are not introduced. Explanation is required on what is the difference?

➔ Thank you for your advice. We added full names of SAR and SIN and an explanation in terms of difference between SAR and SIN modes in P3, L12-15.

*"In this study, we used Synthetic Aperture Radar (SAR) mode, mainly operating on sea ice regions and SAR Interferometric (SIN) mode, mainly operating on steep regions such as margin of ice shelf and ice sheet of level 1b baseline C data, which has 256 and 1024 range bins, respectively (Scagliola, 2014)."*

P3, L9: merge parentheses after PRF.

➔ Thank you for your correction. We merged the parentheses after PRF.

P3, L12: remove "…and this is called multi-looking"

➔ Removed as suggested.

P3, L13: replace "The results in the…" with "Exemplary results

➔ Thank you for your correction. We replaced "the results in the" with "Exemplary results" in P3 L20.

P3, L14: "…with a condition". What is meant? Please rephrase.

➔ We revised the sentence in P3, L20-21.

*"Such waveforms represent temporal distribution of reflected power when the radar pulses reach the surface, describing a flat or rough surface.."*

P4, L21: The data source should be: "These maps are available at the PANGAEA Data Publisher for Earth and Environmental Science data (http://dx.doi.org/10.1594/PANGAEA.854411)."

➔ We corrected the sentence as suggested.

P4, L28: remove "an"

➔ We removed the "an" as suggested.

P5, L7: No URL is required here.

➔ We removed the URL as suggested.

P5, L24-25: Too many references

➔ Two old references were removed.

P6, L3: remove "having"

➔ We removed the word, "having" as suggested.

[revised manuscript text omitted]

---

## Author Comment (AC2) · 11 Jan 2018

**Authors' responses (TC-2017-170)**

The authors would like to thank the editor and the reviewers for their precious time and invaluable comments. The corresponding changes and refinements are highlighted in yellow in the revised paper and are also summarized in our responses below. Authors' responses are in blue. Reviewer's comments are in black. When the manuscript is cited, it is shown in italics.

**Reviewer #2:**

General comments:

This manuscript introduces a new method for Arctic lead detection using a waveform mixture analysis. The method is evaluated by comparison of its performance with other lead detection methods over MODIS imagery. Maps of lead fraction are also presented, and compared with those from other methods. Whilst the application of waveform mixture analysis to CryoSat-2 waveform is novel and I would like to see it published, I think major revisions are needed first.

My key concerns are outlined below, followed by some specific and technical comments.

Thank for your valuable comments. We agree with the reviewer's concerns and comments, and significantly revised the manuscript according to your comments and those from the other reviewers.

The major improvements of the revision are as follows:

1) Clarity of the manuscript significantly improved with more explanation for readers to better understand the context of the paper.

2) We added the discussion for the comparison among the lead detection methods. The evaluation of the lead detection methods was slightly revised throughout the manuscript considering that the validation did not cover the entire Arctic Ocean and the MODIS images for evaluation were partially available.

3) In addition, rescaling beam behavior parameters, pulse peakiness, and backscatter sigma-0 is explained in detail for the transparency of reproduction.

4) Figures were updated and newly added with more clarity.

The authors also would like to thank for the reviewer's minor and technical comments.

Please review our responses below and revised manuscript for details.

Methodology

The method for applying waveform mixture analysis to CryoSat2 waveforms is not clearly explained, and a number of assumptions are not well justified. I suggest that Section 3 requires major revision. Some specific examples:

- On P5 L1-2 the authors state that proper selection of endmembers is essential. At this

point they should explain what is meant by an endmember and endmember vectors, before the linear mixture model is defined. The brief description on P5 L9, which follows model definition, is insufficient.

➔ We added more explanation in section 3.1.

*"An endmember in remote sensing data represents a spectrally pure ground component in a single pixel. For example, it could be pure water, vegetation, bard ground or a soil crust pixel in remote sensing data. Endmembers play the most important role in conducting spectral mixture analysis. Linear spectral mixture analysis assumes that the spectra measured by sensors for a pixel is a linear combination of the spectra of all components within the pixel (Keshava and Mustard. 2002). This technique is widely used to resolve spectral mixture problems in image analysis (Foody and Cox, 1994; Dengsheng et al., 2003; Changshan. 2004; Iordache et al., 2011). Spectral mixture analysis determines the fractions of the components (i.e., classes) found in mixed pixels by producing abundances of the components based on endmembers. The proposed waveform mixture algorithm adopts the concept of spectral mixture analysis. Since the waveform of altimetry within a footprint could be considered to be a mixture of leads and various types of sea ice, spectral mixture analysis can be applied in this framework. In this study, waveforms of CryoSat-2 L1b data were used as endmembers such as the waveform of pure lead and first-year ice (FYI) (Fig. 1). The lead and ice endmembers are used as reference data for separating leads and ice. In order to successfully implement waveform mixture analysis, the proper selection of lead and ice endmembers is essential. "*

● P5 L21: How/Why were these particular CryoSat-2 files selected?

➔ Considering the study period (i.e., Jan.-May, Oct.-Dec. 2011-2016) in this research, a total of 48 orbit files among CryoSate-2 orbit files between 2011 and 2016 were selected to extract endmember samples by month (Jan. to May and Oct. to Dec.), which fully transverse the broad Arctic Ocean (Fig. 2).This explanation is added in P6 L12-14.

*"Among CryoSate-2 orbit files between 2011 and 2016, a total of 48 orbit files were selected to extract endmember samples by month (Jan. to May and Oct. to Dec.), which fully transverse the broad Arctic Ocean (Fig. 2)."*

● P5 L25-28: I do not agree that the comparison of waveforms from March and April 2011-2014 to waveforms over the much broader timeframe (in terms of months and years) of January to May, and October to December 2011-2016 is sufficient to justify the use of DT for waveforms extraction. Ideally the comparison needs to be extended to the full timeframe. Furthermore, the authors should describe how such a comparison

was done.

➔ Thanks for the comment. We added clearer explanation for the comparison. The DT model from Lee et al. (2016) was developed using data (i.e., stack standard deviation, stack skewness, stack kurtosis, pulse peakiness, and backscatter sigma-0) collected in March and April 2011-2014. Thus, the waveforms in other months and years should be compared with the waveforms in March and April 2011-2014 to identify whether the waveforms derived by the DT model during the study period are appropriate to implement the waveform mixture analysis. We added above explanation in the manuscript in P7 L4-7.

*"The DT model from Lee et al. (2016) was developed using data (i.e., stack standard deviation, stack skewness, stack kurtosis, pulse peakiness, and backscatter sigma-0) collected in March and April 2011-2014. Thus, the waveforms in other months and years should be compared with the waveforms in March and April 2011-2014 to identify whether the waveforms derived by the DT model during the study period are appropriate to implement the waveform mixture algorithm."*

➔ It is too many files to completely compare the waveforms between in Mar. and Apr. 2011-2014 and January to May, October to December 2011-2016. We randomly selected CryoSat-2 orbit files on 15th of each month as a large data as possible. The selected waveforms are shown in Response Figs. 1-7. Response Fig. 1 is the waveforms from Mar. and Apr. 2011-2014. Response Figs. 2-7 are the waveforms from and January, February, May, October, November, and December 2011-2016 by year, respectively.

➔ The waveforms are compared through visual inspection. There is no significant difference between them by month or year, which can justify the use of DT to extract endmember candidates. We added this explanation in the manuscript.

[Figure]

**Response Fig. 1** The waveforms in March and April 2011-2014, which is the study period for Lee et al. (2016). CryoSat-2 orbit file names are shown on the top of waveform graphs.

[Figure]

**Response Fig. 2** The waveforms in Jan., Feb., May, Oct., Nov., and Dec 2011. CryoSat-2 orbit file names are shown on the top of waveform graphs.

[Figure]

**Response Fig. 3** The waveforms in Jan., Feb., May, Oct., Nov., and Dec 2012. CryoSat-2 orbit file names are shown on the top of waveform graphs.

[Figure]

**Response Fig. 4** The waveforms in Jan., Feb., May, Oct., Nov., and Dec 2013. CryoSat-2 orbit file names are shown on the top of waveform graphs.

[Figure]

**Response Fig. 5** The waveforms in Jan., Feb., May, Oct., Nov., and Dec 2014. CryoSat-2 orbit file names are shown on the top of waveform graphs.

[Figure]

**Response Fig. 6** The waveforms in Jan., Feb., May, Oct., Nov., and Dec 2015. CryoSat-2 orbit file names are shown on the top of waveform graphs.

[Figure]

**Response Fig. 7** The waveforms in Jan., Feb., May, Oct., Nov., and Dec 2016. CryoSat-2 orbit file names are shown on the top of waveform graphs.

- P6 L25-26: How were the observations permuted (by what increment), and why only 30% of the observations?

  ➔ Thirty percent of the number of lead and ice observations is randomly permuted in a grid for 50 times. We have tested which percentage is appropriate to represent grid sensitivity (Response Fig. 8). As the percentage increases, the grid sensitivity (i.e., standard deviation) also increases but the spatial patterns is almost same. Since the difference is not significant (i.e., 0.05 – 0.1), we considered 30% would be appropriate.

[Figure]

**Response Fig. 8** the grid sensitivity (i.e., standard deviation) test according to the percent (a) 30%, (b) 50%, and (c) 70%.

- P6 L29: The authors should justify why January to April 2011 was chosen as the timeframe for comparison of lead fractions from waveform mixture analysis and existing lead detection methods. This is especially important, as the selection of a different timeframe could alter the results outlined in Section 4.3.

  ➔ We wanted to compare monthly lead fraction maps on a longer time scale. However, Jan.-Mar. 2011 is the only months that monthly lead fraction maps from the other sources are commonly available. In particular, the lead fraction maps from AMSR-E are only available until Mar. 2011 and the lead fraction maps from CryoSat-2 (Wernecke and Kaleschke, 2015) are also available only in Jan. to Mar. 2011.

Evaluation

The evaluation outlined in the manuscript is inconclusive, and I believe the results are oversold. Throughout the manuscript the authors state that evaluation of lead classification with MODIS imagery has shown "better performance" than previous methods (e.g. abstract, conclusions, P7 L19). However, I strongly disagree that this is proven by the results outlined in Section 4.1, Figure 2 or Figure 3. The accuracy of the selection should only be evaluated based on lead statistics (user's and producer's accuracy for leads), and there does not appear to be a significant statistical difference between all methods. Whilst I encourage the authors to discuss

the subtle similarities and differences that they have found between the accuracies of each method, I do not agree with their conclusion. It is stated in the discussion Section 4.1(P13 L4-5) that "the overall accuracy metrics of the proposed waveform mixture analysis approach was comparable to those of the existing methods" which is far more representative of the results shown.

➔ We revised the sentences not to oversell our approach. More comprehensive explanations in terms of accuracies were added in P11. The uncertainty associated with the number of validation images was also discussed. For evaluation, four pairs of cloud free-MODIS images and CryoSat-2 orbit files within the 30-minute difference in the collection time were obtained (section 3.2) considering the cloud interference and polar night. The results are valid for the test dataset, and thus we added the sentences about the uncertainty of the results throughout the manuscript.

*"This lead detection method, based on the waveform mixture analysis, was evaluated with high resolution (250m) MODIS images and showed comparable and promising performance in detecting leads when compared to the previous methods."*

*"Multiple lead classification methods based on CryoSat-2 data were evaluated by visual inspection with high resolution (250m) MODIS images. Leads (i.e., red dot) and sea ice (i.e., light blue dot) are distinguished, depending on the surface condition of lead and sea ice (Fig. 4). For better comparisons, a quantitative assessment is required (Fig. 4). DT from Lee et al. (2016) produced the highest overall accuracy (95.19%), followed by the waveform mixture algorithm (95%), Rose (2013) (93.26%), and Laxon et al. (2013) (91.70%). DT from Lee et al. (2016) produced the highest user's accuracy for leads, while the proposed approach produced the highest producer's accuracy for leads, which implies a slight over-detection of leads by the proposed waveform mixture algorithm. The user's accuracy for leads of Laxon et al. (2013) is the lowest, resulting in much over-detection of leads (i.e., many leads on sea ice; Figs. 4). Similarly, the user's accuracy for ice of Rose (2013) is lower than that of the proposed waveform mixture algorithm, indicating the detection of leads on sea ice, which is shown in Figs. 4b and c. While the performance of the waveform mixture analysis was comparable to the DT algorithm from Lee et al. (2016), the waveform mixture analysis slightly over-estimated leads resulting in a lower user's accuracy than the user's accuracy for leads by DT (Fig. 4 and 5). These are inevitable results because waveforms used in the waveform mixture algorithm are basically extracted by DT from Lee et al. (2016). The lead classification results should be assessed during all the months (i.e., January to May, and October to December) and years (i.e., 2011 to 2016) using MODIS images to thoroughly evaluate the proposed waveform-based algorithm for lead detection. However, the lead classification results in January, February, November, and December were not assessed using MODIS images due to polar nights. Thus, the lead classification results in these months could possibly have uncertainties. It should be also noted that the validation was*

*limited as the MODIS images did not fully cover the entire Arctic region (top in Fig. 4)."*

*"The results show that the proposed approach robustly classified leads with comparable performance to DT from Lee et al. (2016) and slightly better than the existing simple thresholding approaches for lead detection (Rose 2013; Laxon et al., 2013)."*

**Specific comments**

P1 L17: State which parameters (beam behavior, pulse peakiness…)

➔ We added the parameters (i.e., stack standard deviation, stack skewness, stack kurtosis, pulse peakiness, and backscatter sigma) in P1 L18.

P1 L23: Why only mention spring and winter? Leads are common feature in all seasons, and the manuscript covers fall in addition to spring and winter (e.g. Figure 8). If the authors are defining seasons as certain months, then they need to be defined at this point in the text.

➔ Thank you for your comment. We removed the words "spring and winter" in P1 L25.

P1 L23 and L24: "large" and "huge" are meaningless words, without any quantification. This is an issue throughout the manuscript and I suggest the authors check for and remove all such adjectives.

➔ Thank you for your comment. We tried to remove such words without any quantification throughout the manuscript.

P3 L7: Define SAR and SARIn

➔ We added full names of SAR and SIN and an explanation in terms of difference between SAR and SIN modes in P3, L12-15.
*"In this study, we used Synthetic Aperture Radar (SAR) mode, mainly operating on sea ice regions and SAR Interferometric (SIN) mode, mainly operating on steep regions such as margin of ice shelf and ice sheet of level 1b baseline C data, which has 256 and 1024 range bins, respectively (Scagliola, 2014)."*

L7 L6: Are the authors basing this similarity on visual analysis only?

➔ We based this similarity on visual interpretation and waveforms from the literature (Zygmuntowska et al., 2013; Ricker et al., 2015; Lee et al., 2016). Our First-Year Sea ice (FYI) waveform (Fig. 1) is very similar to that of the FYI from literature.

The difference among the waveforms of FYI, MYI, and leads is visually obvious.

P7 L16: A description is needed for how the parameters were rescaled. This is crucial as the authors display these results as being representative of other methods (Rose, 2013; Laxon et al. 2013) after application to Baseline-C CryoSat-2 data, which may not be true. Related to this, it should be made clear that the 'Rose (2013)' and 'Laxon et al. (2013)' results are the authors own version, using the methodologies outlined in the related papers and therefore may differ from the actual results.

➔ Stack standard deviation, stack skewness, stack kurtosis, pulse peakiness, and backscatter sigma-0 were used to identify leads. These parameters in baselines B and C are surely different. The difference is shown in Response Fig. 8. In order to appropriately compare the lead detection methods, we had to rescale the above parameters from baseline C data because Rose (2013), Laxon et al. (2013), and Lee et al. (2016) used these parameters from baseline B. We rescaled the parameters by adding the difference to baseline C data shown in Response Fig. 8. This was also applied to the other months. Therefore, we are sure that the comparison is reasonable. We added additional explanation in the manuscript in P11 L11-12.

*"Since the contrast between the parameters of baselines B and C data is not linear, we rescaled the parameters by adding the difference of the parameters between the two baseline data to baseline C data."*

[Figure]

**Response Fig. 8** The difference between CryoSat-2 baseline B and baseline C of stack standard deviation, pulse peakiness, stack skewness, backscatter sigma-0 in April 2014. This example corresponds to Figure 2b, f, j, and, n in the manuscript.

Figure 2 caption: State day and year of images, in addition to month

➔ We added them in the caption as suggested.

P10 L2: There are no letters on Figures 4 or 5

➔ Thank you for your comment. We removed the letters.

Figure 4 and 5: Whilst I appreciate the overview these figures provide, it is hard to see much detail, such as the higher lead fraction mentioned in spring 2013. There is also no logic in separating into two figures. The data would be better represented by maps for a single year to show seasonal progression of lead fraction, and a time series of mean lead fraction for each month to show inter-annual variability.

➔ We enlarged Figs. 6 and 7 to show more details as suggested.

P12 L8: Again, more transparency needed that these are a reproduction of Laxon et al. (2013) results

➔ Thanks. We clarified this adding more explanation on rescaling data between baselines B and C.

P13 L14: From Figure 2k it appears that the Lee et al. (2016) DT method detects more than one lead in the region. This should be discussed here.

➔ Thank you for your comment. We added a sentence in P20 L13-14.

*"In Lee et al. (2016), DT detected more leads than the other methods, but the validation could not entirely cover the dark area."*

P13 L15: "typical" according to who? If some methods do detect a lead without elevation information how can author can be sure that a lead is not present?

➔ The waveform classified as ice by Laxon et al. (2013) and the waveform mixture algorithm is shown in Response Fig. 9. The waveform appears like FYI waveform than leads through visual inspection. While the lead abundance of the waveform is 0.2294, the ice abundance of the waveform is 0.7706. We rephrased the sentence in P20 L14-15.

*"In fact, since the leads are often refrozen, the shape of the waveforms in that region were likely more similar to the FYI waveform than the lead waveform."*

➔ In order to identify whether the waveform is lead or ice, the spatiotemporal coincident images like MODIS in this study is needed at first. We can empirically identify the leads if satellite or airborne images are unavailable by investigating

beam behavior parameters, pulse peakiness, backscatter sigma-0, and the shape of waveforms.

[Figure]

**Response Fig. 9** The waveform that is detected as ice by Laxon et al. (2013) and the waveform mixture algorithm.

P15 L13: Do not make a claim on trends, for only 6 years of data

➔ Thank you for your comment. We removed the word, "trend".

P15 L14: The months corresponding to spring should be defined in the text, not just the figure caption. Do this at the point in the paper when seasons are first introduced.

➔ Thank you for your comment. We mentioned the specific months in introduction.

Section 4.3. I would appreciate a comment on how these result may been impacted by the timeframe selected by the authors. See also my concerns regarding methodology, final bullet.

➔ We wanted to compare monthly lead fraction maps on a longer time scale. However, Jan.-Mar. 2011 is the only months that monthly lead fraction maps from the other sources are commonly available. The lead fraction maps from AMSR-E are only available until Mar. 2011 and the lead fraction maps from CryoSat-2 (Wernecke and Kaleschke, 2015) are also available only in Jan. to Mar. 2011.

**Technical comments:**

P1 L19: 2011-2016, rather than 2006-2011

P1 L22: "sea ices" to "sea ice"

P1 L27: Change to "… could increase near surface temperature by 3.5K…" or similar

P2 L4: "Recently, **the**Moderate…"

P2 L11: "…of**the**Advanced…"

P2 L13: Change to "Radar altimeters can detect leads as well"

P2 L24: "produced" to "produce"

P3 L3: "… carrying**the**synthetic…"

P3 L3: "burst" to "bursts"

P3 L10: "...**so-called**Doppler beams…" (for clarity)

P11 L7: "…**statistical**uncertainties…"

P14 L1: "Figs. 8" to "Figs. 7"

> ➔ All technical comments were carefully revised. The English of this manuscript was also edited by native professionals.

[revised manuscript text omitted]

---

## Author Comment (AC3) · 11 Jan 2018

**Authors' responses (TC-2017-170)**

The authors would like to thank the editor and the reviewers for their precious time and invaluable comments. The corresponding changes and refinements are highlighted in yellow in the revised paper and are also summarized in our responses below. Authors' responses are in blue. Reviewer's comments are in black. When the manuscript is cited, it is shown in italics.

**Reviewer #3:**

Summary

The paper proposes a novel waveform mixture analysis to detect leads, adopting the concept of linear mixture analysis that is widely used in the field of hyperspectral image analysis. The authors conclude that this method shows a better performance in detecting leads than previous methods. Moreover, spatiotemporal patterns and interannual variability of Arctic-wide lead fractions are discussed.

Thank for your comments. We agreed with your major comments and significantly revised the manuscript according to your comments and those from the other reviewers. We improved insufficient technical explanation for better understanding of readers.

The major improvements of the revision are as follows:

1) We changed confusing words in section 5.3 (i.e., lead dynamics) and abstract. For example, we replaced "investigate" with "briefly examine".

2) We added discussion about the leads that cannot be seen in the MODIS images.

3) We added the discussion for the comparison among the lead detection methods. The evaluation of the lead detection methods was slightly revised throughout the manuscript considering that the validation did not cover the entire Arctic Ocean and the MODIS images for evaluation were partially available.

4) We added the detailed explanation of the N-FINDR algorithm and others. We believed the added explanation will help readers understand the algorithm.

5) Figures were updated and newly added with more clarity.

The authors also would like to thank for the reviewer's minor and technical comments.

Please review our responses below and revised manuscript for details.

General Comments:

The method looks interesting, though I am not really sure if it really outperforms other existing methods. Following the given explanations is not easy and the discussion is sometimes superficial. I try summarize my major concerns:

**Authors' responses (TC-2017-170)**

1) The given results/figures do not sufficiently support the conclusions in the paper. For example, in the Conclusion section, the authors state that "The lead dynamics based on monthly lead fraction maps were examined with Arctic Atmospheric and oceanic circulations". Where is this shown? I am also not sure if the differences between considered algorithms are statistically significant; nor if the evaluation with MODIS images is sufficient, since the resolution is about 250 m, meaning that smaller leads detected with CryoSat-2 might be missed. See also the specific comments below.

➔ We replaced the sentence with "the spatiotemporal distribution of monthly lead fraction maps were documented" in P26 L5. We also replaced the word "investigate" with "briefly examine" in P3 L4.

First of all, although the lateral (i.e., along-track) resolution of CryoSat-2 is ~300m, we do agree that CryoSat-2 could detect smaller leads than 250m. We did not consider the leads that are hardly identifiable in MODIS images and vague leads by cloud interference because we needed reference data for calibrating/validating the lead classification models. Wernecke and Kaleschke. (2015) also used MODIS (250 m) images to validate lead classification using CryoSat-2. They pointed out that the validation is limited because leads smaller than 250m can affect the signal within the footprint. We mentioned an explanation in the manuscript in P7 L26-28.

*"The size of the leads detected by the proposed waveform mixture algorithm is at least 250m or greater because the calibration and validation processes were conducted using MODIS images with 250m spatial resolution. It should be noted that leads smaller than 250m are hardly seen in MODIS images, which implies that there is some uncertainty in the comparison of lead detection methods for small leads."*

Detecting leads smaller than the along track resolution of CryoSat-2 (~300m) with various lead detection methods should be further discussed in detail in future research using high resolution Landsat or SAR imagery. This is quite important in the retrieval of sea ice thickness using an altimeter because leads are used as the tie points for the sea surface height (SSH). The small leads that do not cover the whole footprint do not represent precise SSH. We mentioned the above explanation in the manuscript in P25 L14-19.

*"Detecting leads smaller than the along track resolution of CryoSat-2 (~300m) with various lead detection methods should be discussed in detail later using Landsat or SAR imagery. This is quite important because the leads are used as the tie points for the sea surface height (SSH) in the retrieval of sea ice thickness algorithm."*

➔ We revised the results of comparison of lead classification with more comprehensive explanations in terms of accuracies in P11. The uncertainty associated with the number of validation images was also discussed. For evaluation, four pairs of cloud free-MODIS images and CryoSat-2 orbit files within the 30-minute difference in the collection time were obtained (section 3.2) considering the cloud interference and polar night. The results are valid for the test dataset, and thus we added the sentences about the uncertainty of the results throughout the manuscript.

*"This lead detection method, based on the waveform mixture analysis, was evaluated with high resolution (250m) MODIS images and showed comparable and promising performance in detecting leads when compared to the previous methods."*

*"Multiple lead classification methods based on CryoSat-2 data were evaluated by visual inspection with high resolution (250m) MODIS images. Leads (i.e., red dot) and sea ice (i.e., light blue dot) are distinguished, depending on the surface condition of lead and sea ice (Fig. 4). For better comparisons, a quantitative assessment is required (Fig. 4). DT from Lee et al. (2016) produced the highest overall accuracy (95.19%), followed by the waveform mixture algorithm (95%), Rose (2013) (93.26%), and Laxon et al. (2013) (91.70%). DT from Lee et al. (2016) produced the highest user's accuracy for leads, while the proposed approach produced the highest producer's accuracy for leads, which implies a slight over-detection of leads by the proposed waveform mixture algorithm. The user's accuracy for leads of Laxon et al. (2013) is the lowest, resulting in much over-detection of leads (i.e., many leads on sea ice; Figs. 4). Similarly, the user's accuracy for ice of Rose (2013) is lower than that of the proposed waveform mixture algorithm, indicating the detection of leads on sea ice, which is shown in Figs. 4b and c. While the performance of the waveform mixture analysis was comparable to the DT algorithm from Lee et al. (2016), the waveform mixture analysis slightly over-estimated leads resulting in a lower user's accuracy than the user's accuracy for leads by DT (Fig. 4 and 5). These are inevitable results because waveforms used in the waveform mixture algorithm are basically extracted by DT from Lee et al. (2016). The lead classification results should be assessed during all the months (i.e., January to May, and October to December) and years (i.e., 2011 to 2016) using MODIS images to thoroughly evaluate the proposed waveform-based algorithm for lead detection. However, the lead classification results in January, February, November, and December were not assessed using MODIS images due to polar nights. Thus, the lead classification results in these months could possibly have uncertainties. It should be also noted that the validation was limited as the MODIS images did not fully cover the entire Arctic region (top in Fig. 4)."*

*"The results show that the proposed approach robustly classified leads with comparable performance to DT from Lee et al. (2016) and slightly better than the existing simple thresholding approaches for lead detection (Rose 2013; Laxon et al., 2013)."*

2) The method description in the paper lacks more detailed information in the methodical part. The methods/algorithms are explained very briefly, e.g. "N-FINDER", "MATLAB toolbox for linear unmixing with the interior point least square algorithm". Although the authors refer to literature sometimes, these methods should be explained more detail, since they are essential for understanding the study. Specifically, in the beginning, it should be explained what "endmembers" are and what they represent. The same applies to the "abundance fraction". Moreover, the authors do not show explicitly how the abundance fraction is derived. Additional figures explaining intermediate steps would be very helpful for understanding.

➜ Thank you for your advice. First of all, we added more the detailed explanation of the N-FINDR algorithm in P6 L20-30 and P7 P1-3. Secondly, we replaced "linear unmixing with the interior point least square algorithm" (Chouzenoux et al., 2014) with "basic linear spectral mixture analysis" described in the manuscript because the former method is too slow to implement the waveform mixture algorithm. There are about more than 700 CryoSat-2 orbit files on average in a month. It takes too long to run the waveform mixture algorithm for all single CryoSat-2 orbit files. Therefore, we applied the concept of the basic linear spectral mixture analysis to the waveform of CryoSat-2 to reduce running time without significant change in the results of lead and ice abundances as well as monthly lead fractions. Thirdly, we introduced "endmembers" as early as possible in the beginning of section 3.1. Lastly, how the abundance is derived is described in P5 L27. We believe the additional explanation will help readers understand the process used in the study.

*"The lead and sea ice endmembers (i.e., the most representative waveforms) are a key factor in the successful implementation of the waveform mixture algorithm. In order to avoid the subjective selection of endmembers, a number of endmember candidates were extracted by the DT algorithm (Lee et al., 2016) and the N-FINDR algorithm determined the optimum lead and ice endmembers. The N-FINDR algorithm basically uses the fact that the N spectral dimension and the N-volume (V), defined by a simplex with pure pixels, are always greater than any other combinations (Winter 1999). It operates by inflating a simplex inside of the data (endmembers), starting with any pixel set. The endmember is replaced with another endmember, and the volume is recalculated. The endmember is replaced with the spectrum of the new pixel if the volume increases. This process repeats until the volume does not increase (i.e., until there is no replacement).*

*The Volume (V) of the simplex containing synthetic endmember sets is proportional to the determinant. This algorithm has been widely used for automatically selecting representative endmembers (Winter, 1999; Zortea and Plaza, 2009; Erturk and plaza, 2015; Ji et al., 2015; Chi et al., 2016)."*

3) The selection of the endmembers need more explanation. Where are the 48 collected CryoSat-2 orbits located? A map would be very helpful here. Also, I wonder how this approach deals with different ice types, given that first-year ice waveforms are different from multiyear ice waveforms? Reading section 3.1, it sounds like only first-year ice waveforms have been considered? The next issue is the nonlinear mixing as mentioned correctly. Due to the specular reflection, a lead (of a certain size) will always dominate the waveform. This is even more the case for the Doppler SAR processed waveforms. How is this handled in the WMA?

➜ We added Response Fig. 1 showing 48 CryoSat-2 orbits (Fig. 2 in the manuscript) and agree with that this figure would be helpful to understand the selection of the endmembers.

[Figure]

**Response Fig. 1** The 48 CryoSat-2 orbit files from Jan. 2011 to Dec. 2016 used for extraction of endmember waveforms. The selected CryoSat-2 orbit files relatively cover the entire Arctic Ocean.

➔ The endmember of first-year ice (FYI) waveform likely covers multi-year ice (MYI) waveform. This is because the shape of MYI waveform is obviously different from lead waveform, and a bit similar to FYI waveform. Response Fig. 2 shows the examples of typical MYI waveforms, describing rough sea ice surface with high power in trailing edge (Ricker et al., 2015; Zygmuntowska et al., 2013). Since we considered binary classification between leads and ice, the ice abundance for MYI waveforms make it easier to classify the waveforms as ice, which is shown in Response Fig. 2.

[Figure]

**Fig. 2** The examples of multi-year sea ice waveforms from CryoSat-2 orbit files in 25 May 2014.

➡ CryoSat-2 (i.e., SAR altimeter) can represent small lead in the sea ice (<300m). The

lead like waveforms depends on the size of lead within the footprint. However, we are not sure that how different lead sizes (< 300m) affect the waveform. We think that the waveforms from CryoSat-2 prone to be more sensitive to the specular reflection of the leads than the diffuse reflection of sea ice as we documented in P6 L4. The waveform mixture algorithm might tend to overestimate leads. This is confirmed from the classification results in section 4.1

➜ How the leads smaller than the along track resolution of CryoSat-2 (~300m) affect waveform should be further studied using high spatial resolution Landsat or SAR images. This is documented in P25 L18-21.

4) One of the objectives is to "investigate the relationship between Arctic lead fraction and thermodynamics and ice dynamics". However, this is discussed just very briefly in Section 5.3. As mentioned above, a thorough examination of the linkage to atmospheric forcing and ice dynamics is not shown.

➜ We replaced the world "investigate" with "briefly examine" in P3 L4.

The issues, listed above, should be addressed by the authors. Moreover, some sentences are unclear or imprecise (see specific comments). Taken together, these omissions means that major revisions are needed.

Specific Comments:

P1L22. "sea ices" – the plural of sea ice sound odd.

➜ Thank you for your correction. We corrected the word.

P2L27: "could make near surface temperature up to a 3.5 K"- this sentence is confusing and should be rewritten. I suppose you mean that an increase in lead fraction leads to an increase in near-surface temperature of up to 3.5 K?

➜ Thank you for your comment. We corrected the sentence in P1 L28-29.

*"Lüpkes et al. (2008) showed that a 1% change in sea ice concentration owing to an increase of lead fraction could increase near surface temperature in the Arctic by 3.5 K."*

P3L5: "CryoSat-2 takes an advantage of SIRAL to detect smaller leads (e.g., ~300m)" – why should the lead size related the Doppler beam footprint (300m)? The actual size of the lead might be smaller, since the specular return from the mirror-like lead surface will dominate the

waveform, also if the illuminated surface is a mixture of sea ice and lead.

➔ Thank you for your comment. We agree with your opinion. The leads smaller than ~300m are captured by the waveforms from CryoSat-2 data. We removed the CryoSat-2 footprint in the sentence in P3 L10.

P3 Section 2.1: It should be clearly written which data are used here (I suppose level 1b). Which Baseline has been used (C?)? Which period is considered?

➔ We used CryoSat-2 baseline C data mentioned in P3 L14-15. The period of CryoSat-2 level 1b baseline C data used in this study is in Jan. – May, Oct. – Dec. 2011-2016.

P3 Section 2.2: Same as above: Which data product version has beed used? Which period?

➔ We used CryoSat-2 baseline C data mentioned in P3 L14-15. The period of CryoSat-2 level 1b baseline C data used in this study is in Jan. – May, Oct. – Dec. 2011-2016.

P5L5: "vectors" -> vector

➔ Corrected as suggested.

P5L21: What do you mean with "CryoSat-2 files"? An orbit file?

➔ Yes. We added the "orbit" in the sentence. Fig. 2 in the manuscript would be helpful for readers understand the CryoSat-2 orbit files used in the study.

P5L25: "Waveforms from March and April between 2011 and 2014 were compared to those from January to May, and October to December between 2011 and 2016 (not shown), resulting in little difference between them"- Why do you separate between the two periods (January-May, October-December)?

➔ Jun. to Sep. is generally considered as the melting season. In this season, the presence of leads as well as meltpond in sea ice are dominant. It is difficult to accurately distinguish leads from sea ice due to the fact that the waveform of the meltpond is quite similar to that of leads. Since the lead detection methods for the retrieval of sea ice thickness do not work well in the melting season, the sea ice thickness in the melting season is still unavailable (Tilling et al., 2017). More advanced lead detection methods need to be developed for the summer season with careful consideration of meltpond.

P7L18-26: The MODIS resolution is about 250 m. What about smaller leads (<250m)? Due to

their specular surface, they could be detected by CryoSat-2, but not with MODIS. Therefore, I wonder how representative this evaluation is?

➔ First of all, although the lateral (i.e., along-track) resolution of CryoSat-2 is ~300m, we do agree that CryoSat-2 could detect smaller leads than 250m. We did not consider the leads that are hardly identifiable in MODIS images and vague leads by cloud interference because we need reference data for calibration and validation of lead classification. Wernecke and Kaleschke. (2015) also used MODIS (250 m) images to validate lead classification using CryoSat-2. They pointed out that the validation is limited because lead smaller than 250m can affect the signal within the footprint. We mentioned the above explanation in the manuscript in P7 L26-29.

*"The size of the leads detected by the proposed waveform mixture algorithm is at least 250m or greater because the calibration and validation processes were conducted using MODIS images with 250m spatial resolution. It should be noted that leads smaller than 250m are hardly seen in MODIS images, which implies that there is some uncertainty in the comparison of lead detection methods for small leads."*

➔ Detecting leads smaller than the along track resolution of CryoSat-2 (~300m) with various lead detection methods should be further discussed in detail in future research using high resolution Landsat or SAR imagery. This is quite important in the retrieval of sea ice thickness using an altiemter because the leads are used as the tie points for the sea surface height (SSH). The small leads that do not cover the whole footprint do not represent precise SSH. We mentioned the above explanation in the manuscript in P25 L14-17.

*"Detecting leads smaller than the along track resolution of CryoSat-2 (~300m) with various lead detection methods should be discussed in detail later using Landsat or SAR imagery. This is quite important because the leads are used as the tie points for the sea surface height (SSH) in the retrieval of sea ice thickness algorithm."*

➔ We revised the results of comparison of lead classification with more comprehensive explanations in terms of accuracies in P11. The uncertainty associated with the number of validation images was also discussed. For evaluation, four pairs of cloud free-MODIS images and CryoSat-2 orbit files within the 30-minute difference in the collection time were obtained (section 3.2) considering the cloud interference and polar night. The results are valid for the test dataset, and thus we added the sentences about the uncertainty of the results throughout the manuscript.

*"This lead detection method, based on the waveform mixture analysis, was evaluated with high resolution (250m) MODIS images and showed comparable and promising performance in detecting leads when compared to the previous methods."*

*"Multiple lead classification methods based on CryoSat-2 data were evaluated by visual inspection with high resolution (250m) MODIS images. Leads (i.e., red dot) and sea ice (i.e., light blue dot) are distinguished, depending on the surface condition of*

*lead and sea ice (Fig. 4). For better comparisons, a quantitative assessment is required (Fig. 4). DT from Lee et al. (2016) produced the highest overall accuracy (95.19%), followed by the waveform mixture algorithm (95%), Rose (2013) (93.26%), and Laxon et al. (2013) (91.70%). DT from Lee et al. (2016) produced the highest user's accuracy for leads, while the proposed approach produced the highest producer's accuracy for leads, which implies a slight over-detection of leads by the proposed waveform mixture algorithm. The user's accuracy for leads of Laxon et al. (2013) is the lowest, resulting in much over-detection of leads (i.e., many leads on sea ice; Figs. 4). Similarly, the user's accuracy for ice of Rose (2013) is lower than that of the proposed waveform mixture algorithm, indicating the detection of leads on sea ice, which is shown in Figs. 4b and c. While the performance of the waveform mixture analysis was comparable to the DT algorithm from Lee et al. (2016), the waveform mixture analysis slightly over-estimated leads resulting in a lower user's accuracy than the user's accuracy for leads by DT (Fig. 4 and 5). These are inevitable results because waveforms used in the waveform mixture algorithm are basically extracted by DT from Lee et al. (2016). The lead classification results should be assessed during all the months (i.e., January to May, and October to December) and years (i.e., 2011 to 2016) using MODIS images to thoroughly evaluate the proposed waveform-based algorithm for lead detection. However, the lead classification results in January, February, November, and December were not assessed using MODIS images due to polar nights. Thus, the lead classification results in these months could possibly have uncertainties. It should be also noted that the validation was limited as the MODIS images did not fully cover the entire Arctic region (top in Fig. 4)."*

*"The results show that the proposed approach robustly classified leads with comparable performance to DT from Lee et al. (2016) and slightly better than the existing simple thresholding approaches for lead detection (Rose 2013; Laxon et al., 2013)."*

Figure 2: An overview map with the locations of the MODIS images would be helpful.

➔ Thank you for your comment. We added an overview map with locations of the cropped MODIS images in Fig. 2 in the manuscript.

[Figure]

**Response Fig. 3** Visual comparison of lead classifications: (a) – (d) lead classifications based on Rose (2013), (e) – (h) lead classifications based on Laxon et al. (2013), (i) – (l) lead classifications based on decision trees from Lee et al. (2016), and (m) – (p) lead classifications based on the proposed waveform mixture analysis. The MODIS data were collected on 27 March 2016 (a, e, i, and m), 17 April 2014 (b, f, j, and n), 25 May 2015 (c, g, k, and o), and 10 October 2015 (d, h, l, and p). An overview map of the location of cropped MOIDS images is in top of the figure.

Figure 3: Unit is missing. Percentage?

➜ We added the unit (percentage) in Fig. 4 in the manuscript.

[Figure]

*Unit: Percentage (%)

**Response Fig. 5** Accuracy assessment results for lead detection by method—three existing methods and the proposed waveform mixture analysis (WMA).

P9L7: "The Areas around the coast line"- To me it seems that lead fractions is higher at the ice edge and in the marginal ice zones, like Barents Sea?!

➜ Thank you for your comment. We also think that "marginal ice zones" is more appropriate than "coast line" in P15 L9-10.

*"The areas in the marginal ice zones line of the Arctic Ocean clearly show high lead fraction due to the shear zone (i.e., an area of deformed sea ice along the coast, Serreze and Barry, 2005) and outflow of sea ice."*

P11L10-12: I would argue that this conclusion is not valid: You average over one month, so leads are also propagating, opening and closing during the period. Certainty, when you have large ice drift, like in the Beaufort Gyre. Therefore, the standard deviation might not be reflect the uncertainty here.

➔ This is also an issue for calculating monthly lead fraction maps using satellite altimetry data. It should also be noted that it is hard for the altimeter-based lead detection methods used in such as Wernecke and Kaleschke (2015) and this study to identify the propagating, opening, closing of leads because sea ice and leads generally move when the altimeters revisit a certain grid. In this study, a monthly lead fraction was derived by dividing the number of lead observations with total observations within a 10km grid in a month. The monthly lead fraction strongly depends on the number of lead and ice observations in the grid. However, while there are more than 30 CryoSat-2 observations in the 10 km grid around centre of the Arctic, the number of CryoSat-2 observations less than 5 are in the 10 km grid around the coastline of Arctic Ocean. In the section 4.3 and Fig. 8 in the manuscript, we try to identify how the lead fraction in the grid change by randomly and repetitively changing the thirty percent of the number of leads and ice in the grid. In a grid with a small number of lead observations, the lead fraction significantly changes even by a small change in the number of the lead observations. In this context, the high standard deviation from this sensitivity analysis is likely to represent high sensitivity of the lead fraction in the grid. Above explanation is added in section 3.3 of the manuscript. Furthermore, we changed the word "uncertainty" to "sensitivity". We added the explanation of drawbacks for the monthly lead fraction maps derived using altimeter data in the section 3.2.

*"It also should be noted that it is hard for the altimeter-based lead detection methods used in such as Wernecke and Kaleschke (2015) and this study to identify the propagating, opening, closing of leads because sea ice and leads generally move when the altimeters revisit a certain grid."*

Figure 6 Unit is missing. Percentage?

➔ In Figure 8a-h, the unit is the number of lead and ice observations. There is no unit in Figure 6i-l because it is just the standard deviation of 50 lead fraction (i.e., scaled) derived by 50 iterations. The high value in Figure 8i-l means that it is sensitive to the change in the number of lead and ice observations.

Figure 6: Why do you get these orbit patterns in the sea ice fraction maps, certainly in February 2011?

➔ The orbit patterns were attributed by the large number of CryoSat-2 orbit files in Feb. 2011. Too many wrong orbit files in Feb. 2011 were downloaded and used to create the

sea ice fraction maps. We re-downloaded all of the orbit files and re-generated the maps, which do not have such orbit patterns in the sea ice fraction maps.

P17L3-5: "In addition, this study showed the high inter-annual variability of Pan-Arctic lead fractions in recent years (i.e., 2011-2016), which implies that sea ice becomes more vulnerable to atmospheric and oceanic forcing." How does the interannual variability of lead fractions imply that sea ice becomes more vulnerable to atmospheric and oceanic forcing? This is not clear to me.

➔ The sea ice cover over the Arctic Ocean has continuously diminished by global warming and Arctic amplification. There is a linear diminishing trend of sea ice extent in the Arctic for a long time frame (e.g., 1978 - present). However, the interannual variability of recent sea ice extent is strong by recent atmospheric and oceanic anomalies (Tilling et al., 2015; Kim et al., 2017; Ricker et al., 2017). For clarity, we revised the sentence in P25 L8-10.

[revised manuscript text omitted]

---

## Referee Report (RR1)

The authors have worked hard to address my original comments. The paper is much improved as:

- The method is expanded and better structured
- The authors no longer oversell the evaluation results from comparing different lead classification methods with MODIS imagery. For example, I now agree with the abstract statement that waveform mixture analysis shows "comparable and promising performance" compared with other lead detection methods, rather than the "better performance" stated before.
- Display items are clear

However, I have a few remaining comments that should be addressed. Please see below.

P6 L12-14: The highlighted addition still doesn't explain which particular orbit files are selected. First of the month? A consistent date each time?

P7 L6: "…months and years should be compared **using visual analysis** with the waveforms… i.e. introduce the method of comparison here rather than L9.

P7 L9: The authors should comment on the limitations of such a visual analysis, rather than a statistical one, to compare waveforms.

P10 L10-13: Please explain in the manuscript that as the percentage of permuted observations increases, the grid sensitivity also increases but the difference is not significant, hence 30% was chosen.

---

## Author Response (AR2)

**Authors' responses (TC-2017-170)**

The authors would like to thank the editor and the reviewers for their precious time and invaluable comments. The corresponding changes and refinements are highlighted in yellow in the revised paper and are also summarized in our responses below. Authors' responses are in blue. Reviewer's comments are in black. When the manuscript is cited, it is shown in italics.

Editor's comments

I strongly agree with referee #3 comment that the discussion on the relationship between Arctic lead fraction, thermodynamics and ice dynamics is too superficial (this point was also raised by referee #1's report on the initial submission). I suggest to either remove section 5.3 (possible integrating some of the text in other parts of the manuscript) or provide a more thorough analysis. In case of the latter, I might contact an additional expert in this field to comment on your analysis

➔ Thank you for your comment. As suggested, we removed section 5.3 and moved some of the text to section 4.2 (L14 P21-28).

*"The Arctic Ocean circulations have contributed to the change in the state of sea ice. The lead fraction in Northwestern Greenland in Figs. 6 and 7 is low because of the convergence of sea ice by two major circulations, which was clearly shown in Kwok (2015). Kwok et al. (2013) revealed that the currents speed of Beaufort Gyre and Transpolar Drift increased from the years of 1982 to 2009 and this makes the fraction of multi-year ice decrease. However, the increasing lead fraction from the years of 2011 to 2016 in this study was not seen due to the high inter-annual variability of a lead fraction, particularly in the spring season (Figs. 6 and 7). High sensitivities in the marginal sea ice zone might result in not catching the increasing trend of Arctic lead fraction shown in the literature. In order to properly compare the Arctic current circulations and lead fraction, long-term lead fraction data are needed."*

➔ We also changed the third objective in the introduction (P2 L31-32), accordingly.

From "briefly examine the relationship between Arctic lead fraction and thermodynamics and ice dynamics."

To *"3) examine the spatiotemporal distribution of lead fractions."*

In addition to the suggestions of the referees, I ask you to make the following changes:

1. On p.25 remove "which implies that recent ice status has become more vulnerable to anomalous atmospheric and oceanic conditions." Given the short time span of the observations, this is speculative.

➔ Thank you for your comment. We revised the sentence (P23 L10-11).

*"In addition, this study showed the high inter-annual variability of Pan-Arctic lead fractions in recent years (i.e., 2011-2016)."*

2. In section 5.4, on novelty and limitations, justify why you present method should be used rather than the method proposed Lee et al., 2016.

➔ Thank you for your comment. We added a brief justification of the use of the waveform mixture algorithm (P23 L8-10).

*"The proposed waveform mixture algorithm would be very useful in an operational system than the threshold-based methods including Lee et al. (2016)."*

3. Include a discussion on the difficulties in detection leads in the summer season, as in your response to R3 on the P5L25 comment.

➔ We added the discussion (P14 L3-7).

*"The period from June to September is generally considered as the melting season. In this season, the presence of leads as well as melt pond in sea ice are dominant. It is difficult to accurately distinguish leads from sea ice due to the fact that waveform of the melt pond is quite similar to that of leads. Since the lead detection methods for the retrieval of sea ice thickness do not work well in the melting season, the sea ice thickness during the melting season is still unavailable (Tilling et al., 2017)."*

Technical correction:

Rephrase "CryoSat-2 observations less than 5 are in the 10 km grid around the coastline of Arctic" to, for example, "less than 5 observations are found in the 10 km grids in the marginal zones of the Arctic ocean"

➔ We revised the sentence as suggested.

The authors would like to thank the editor and the reviewers for their precious time and invaluable comments. The corresponding changes and refinements are highlighted in yellow in the revised paper and are also summarized in our responses below. Authors' responses are in blue. Reviewer's comments are in black. When the manuscript is cited, it is shown in italics.

**Reviewer #2:**

The authors have worked hard to address my original comments. The paper is much improved as:

- The method is expanded and better structured

- The authors no longer oversell the evaluation results from comparing different lead classification methods with MODIS imagery. For example, I now agree with the abstract statement that waveform mixture analysis "comparable and promising performance: compared with other lead detection methods, rather than the "better performance" stated before.

- Display items are clear

However, I have a few remaining comments that should be addressed. Please see below.

P6 L12-14: The highlighted addition still doesn't explain which particular orbit files are selected. First of the month? A consistent date each time?

➔ The CryoSat-2 orbit files were selected on the 15$^{th}$ of each month 2011-2016 (P6 L14).

*"The selection of endmembers is essential in the framework of waveform mixture analysis. Among CryoSat-2 orbit files between 2011 and 2016, a total of 48 orbit files were selected to extract endmember samples by month (15$^{th}$ from Jan. to May and from Oct. to Dec.), which fully transverse the broad Arctic Ocean (Fig. 2)."*

P7 L6: "…months and years should be compared **using visual analysis** with the waveforms… i.e. introduce the method of comparison here rather than L9.

➔ Thank you for your comment. We added "through visual analysis" (P7 L7) and deleted the phrase from the original location as suggested.

P7 L9: The authors should comment on the limitations of such a visual analysis, rather than a statistical one, to compare waveforms.

➔ Thank you for your comment. We commented on the limtations of such visual analysis in P7 L12-13.

*"However, such visual analysis cannot guarantee how the waveforms are quantitatively different by month and year."*

P10 L10-13: Please explain in the manuscript that as the percentage of permuted observations increases, the grid sensitivity also increases but the difference is not significant, hence 30% was chosen.

**Authors' responses (TC-2017-170)**

➜ Thank you for your comment. We added the explanation in P10 L10-12.

*"We tested various percentage values to identify which percentage is appropriate to represent grid sensitivity. As the percentage increased, the grid sensitivity (i.e., standard deviation) also increased but the spatial difference was not significant, hence a 30 % was chosen."*

**Authors' responses (TC-2017-170)**

The authors would like to thank the editor and the reviewers for their precious time and invaluable comments. The corresponding changes and refinements are highlighted in yellow in the revised paper and are also summarized in our responses below. Authors' responses are in blue. Reviewer's comments are in black. When the manuscript is cited, it is shown in italics.

**Reviewer #3:**

General Comments:

I acknowledge the efforts of the authors to revise the manuscript according to my comments. Some of my major concerns have been addressed sufficiently from my point of view, especially the lack of detailed information in some of the sections and certainly regarding the WMA algorithm.

However, two major concerns remain:

1) To me, it seems that the proposed waveform mixture algorithm does not represent a significant improvement compared to former lead detection algorithms methods, also given the limitations of the evaluation due to the resolution of the MODIS images. This is shown by the evaluation shown in Figure 5. The overall accuracy seems to be even slightly lower than in the method proposed by the author in a previous paper (Lee et al., 2016). Except the potential independence of CryoSat baseline changes, the authors cannot convincingly show why one should use the WMA algorithm instead of former used algorithms. However, since the method itself used here is novel, this might merit publication.

➔ The differences of the lead user's accuracy and overall accuracy between Lee et al., (2016) and the waveform mixture algorithm are 2.61% and 0.19 %. We agree with your point but we believe that the waveform mixture algorithm will be worthwhile in an operational system because the proposed approach does not require to change any parameters when CryoSat-2 baseline is updated.

2) Objective 3) is still treated superficial. Although in the revision you now write "briefly examine", it is still remains an objective. But what you show is rather a discussion of the state of the art. I am missing a detailed analysis. It is also now shown how the forming of leads is related to atmospheric forcing. On the other hand, this objective is not mentioned neither in the title nor in the abstract. Therefore, I would suggest to either expand this analysis in this paper, e.g. considering atmospheric circulation patterns, or to do it in a different study.

➔ Thank you for your comment. According to your comment and Editor's, we removed section 5.3 and moved some relevant text to section 4.2 (L14 P21-28). We also revised the third objective, accordingly.

From "briefly examine the relationship between Arctic lead fraction and thermodynamics and ice dynamics."

To *"3) examine the spatiotemporal distribution of lead fractions."*

(L14 P21-28). *"The Arctic Ocean circulations have contributed to the change in the state of sea ice. The lead fraction in Northwestern Greenland in Figs. 6 and 7 is low because of the convergence of sea ice by two major circulations, which was clearly shown in Kwok (2015). Kwok et al. (2013) revealed that the currents speed of Beaufort Gyre and Transpolar Drift increased from the years of 1982 to 2009 and this makes the fraction of multi-year ice decrease. However, the increasing lead fraction from the years of 2011 to 2016 in this study was not seen due to the high inter-annual variability of a lead fraction, particularly in the spring season (Figs. 6 and 7). High sensitivies in the marginal sea ice zone might result in not catching the increasing trend of Arctic lead fraction shown in the literature. In order to properly compare the Arctic current circulations and lead fraction, long-term lead fraction data are needed."*

In general, I suggest to streamline the manuscript to improve readability, see therefore some comments below.

Specific Comments:

P1 L10-12: I suggest to delete these first two sentences. They rather belong to the Introduction.

➔ As suggested, we removed the two sentences in the abstract.

P1 L15: delete "based on the waveform mixture analysis" – repetition.

➔ We removed the phrase.

P1 L20-21: "which show a strong inter-annual variability of recent sea ice cover during 2011-2016, excluding the summer season (i.e., June to September)" – Can you quantify this? What means "strong" in this context? Otherwise this sentence is not really valuable.

➔ The inter-annual variability of averaged lead fraction is identified in the revised manuscript (Fig. 8). We removed the word, "strong" from the abstract and conclusion.

P3 L1: "is not easily affected" – What does it mean? How is it affected? Be more precise here. You might consider to shift that part to the discussion.

➔ Thank you for your comment. We added more explanation (P2 L 28-30).

*"The lead detection using the proposed waveform mixture algorithm does not need to change any parameters to detect leads when the CryoSat-2 baseline is updated, which is a significant advantage compared to the existing threshold-based lead detection methods."*

P3 L15: "The period of CryoSat-2 level 1b baseline C data in this study is in Jan. – May, Oct. – Dec. 2011-2016" – Why are you using data in May also? Substantial surface melt can occur already in some regions. This will surely affect the dielectric properties of the (snow) surface

and therefore will affect the sea ice waveforms. Wouldn't that bias also the end-member selection?

➔ Thank you for your comment. We have checked the waveforms in May. Although surface melting occurs on the sea ice surface, the DT can select appropriate candidates for lead and ice waveform endmembers. Response Figs. 1 and 2 show that there is no significant difference in the waveforms extracted between in May and other months (Reseponse Fig . 3). Therefore, we believe that the waveforms in May would not make a bias for endmember selection. Although we could not check all of the candidates of lead and ice waveform endmembers in May 2012-2016, a number of the waveforms randomly selected from the candidates in May suggest that the endmember selection seems appropriate based on lead and ice endmembers shown in Fig. 1 in the revised manuscript and response Fig. 3.

[Figure]

**Response Fig. 1** Examples of the lead endmember candidates selected by the DT in May 2012-2016. The text above each figure is the CryoSat-2 file name.

[Figure]

**Response Fig. 2** Examples of the ice endmember candidates selected by the DT in May 2012-2016. The text above each figure is the CryoSat-2 file name.

[Figure]

**Response Fig. 3** Examples of the ice endmember candidates selected by the DT in January, February, March, April, November, and December 2012-2016. The text above each figure is the CryoSat-2 file name.

P15 L15/16: "the lead fraction starts to increase from April. This indicates an increasing lead fraction…" – Please rephrase.

➔ We rephrased the sentence (P14 L19-20).

*"While the lead fraction decreases from October to March (i.e., freezing season) with the minimum in March, the lead fraction starts to increase from April."*

P26 L5: "The spatiotemporal distribution of monthly lead fraction maps were documented." – That's not really a conclusion.

➔ We removed the sentence in the conclusion.

P26 L6/7: "Unlike thresholds based lead detection methods, the waveform mixture analysis is less influenced on the update of baseline version of CryoSat-2 data" – Again, please, be a bit more specific here. What does "less" mean here. Indeed, this would be a unique feature probably.

➔ Thank you for your comment. We rephrased it with specific explanation (P23 L5-7).

*"Unlike thresholds based lead detection methods, since the waveform mixture algorithm solely uses waveforms, not beam behaviour parameters, it does not need 
[revised manuscript text omitted]

On the other hand, the waveform mixture analysis depends on the quality of the endmembers. Although the use of the N-FINDR algorithm decreased the subjective selection of endmembers, waveform samples of leads and sea ice derived by DT algorithm from Lee et al. (2016) may introduce uncertainty because the algorithm was validated for March and April from 2011 to 2014. The leads that are not identifiable in the MODIS images were not considered in this study. Detecting leads smaller than the along track resolution of CryoSat-2 (~300m) with various lead detection methods should be further discussed in detail in future research using high resolution Landsat or SAR imagery. This is quite important in the retrieval of sea ice thickness using an altimeter because leads are used as the tie points for the sea surface height (SSH). For example, how the leads smaller than the along-track resolution of CryoSat-2 affect the waveform and SSH should be further investigated. The spatial resolution of monthly lead fraction maps improved up to 10 km, showing a detailed spatial distribution of leads in the Arctic. For example, 10km lead fractions showed significant variations in some regions, while 50 km or 100km lead fractions did not because lead fractions are averaged, resulting in blurred spatial patterns.

**6. Conclusions**

The waveform mixture algorithm was proposed to detect leads with CryoSat-2 L1b data. The lead and sea ice waveforms were considered as endmembers that are essential to implement waveform mixture algorithm. The endmembers (i.e., representative waveforms of leads and sea ice) were extracted by the N-FINDR algorithm among numerous waveforms (i.e., 420,858 waveforms of sea ice and 8,501 waveforms of leads). The thresholds to make a binary classification were determined by calibrating lead and sea ice abundances with reference data extracted from a high resolution (250m) MODIS images. The results show that the proposed approach robustly classified leads with comparable performance to DT from Lee et al. (2016) and slightly better than the existing simple thresholding approaches for lead detection (Rose 2013; Laxon et al., 2013). Furthermore, the lead detection of waveform mixture algorithm was comparable to the DT based lead detection

method (Lee et al., 2016), suggesting a sea ice freeboard can be retrieved with the robust lead detection method using waveform mixture analysis. Monthly lead fraction maps were produced using the proposed waveform mixture approach, showing clear inter-annual variability. The results of the lead fraction maps are consistent with the findings of recent studies (Tilling et al., 2015; Ricker et al., 2017; Kim et al., 2017).

5     Unlike the threshold-based lead detection methods, since the waveform mixture algorithm solely uses waveforms, not beam behaviour parameters, it does not need 
[revised manuscript text omitted]

---

## Author Response (AR4)

The authors would like to thank the editor and the reviewers for their precious time and invaluable comments. The corresponding changes and refinements are highlighted in yellow in the revised paper and are also summarized in our responses below. Authors' responses are in blue. Reviewer's comments are in black. When the manuscript is cited, it is shown in italics.

Thanks for editor's comments. The authors also thank for Editor's contribution to the significant improvement of the manuscript.

**Editor's comments:**

I have reviewed your revised manuscript and it's my pleasure to inform you that your work is now almost ready for publication in The Cryosphere. I would like to make one textual change in the following sentence in section 4.2:

"High grid sensitivities (referring to section 4.3) of lead fraction in the marginal sea ice zone due to the smaller number of observations than in the higher latitudes might not represent the increasing trend of Arctic lead fraction shown in the literature."

Grid sensitivities are discussed in the next section. For the flow of the paper, it would be better to incorporate this statement to section 4.3:

[revised manuscript text omitted]